# NCLX prevents cell death during adrenergic activation of the brown adipose tissue

Essam A. Assali [1,2,3,4], Anthony E. Jones [3], Michaela Veliova[1,3], Rebeca Acín-Pérez[1,3], Mahmoud Taha[4], Nathanael Miller[1,3], Michaël Shum[1], Marcus F. Oliveira[5], Guy Las[2], Marc Liesa [1,3], Israel Sekler[4✉] & Orian S. Shirihai [1,2,3✉]

A sharp increase in mitochondrial $Ca^{2+}$ marks the activation of brown adipose tissue (BAT) thermogenesis, yet the mechanisms preventing $Ca^{2+}$ deleterious effects are poorly understood. Here, we show that adrenergic stimulation of BAT activates a PKA-dependent mitochondrial $Ca^{2+}$ extrusion via the mitochondrial $Na^+/Ca^{2+}$ exchanger, NCLX. Adrenergic stimulation of NCLX-null brown adipocytes (BA) induces a profound mitochondrial $Ca^{2+}$ overload and impaired uncoupled respiration. Core body temperature, PET imaging of glucose uptake and $VO_2$ measurements confirm a thermogenic defect in NCLX-null mice. We show that $Ca^{2+}$ overload induced by adrenergic stimulation of NCLX-null BAT, triggers the mitochondrial permeability transition pore (mPTP) opening, leading to a remarkable mitochondrial swelling and cell death. Treatment with mPTP inhibitors rescue mitochondrial function and thermogenesis in NCLX-null BAT, while calcium overload persists. Our findings identify a key pathway through which BA evade apoptosis during adrenergic stimulation of uncoupling. NCLX deletion transforms the adrenergic pathway responsible for thermogenesis activation into a death pathway.

[1] Division of Endocrinology, Department of Medicine, David Geffen School of Medicine, University of California Los Angeles, Los Angeles, CA 90095, USA. [2] Department of Clinical Biochemistry, Faculty of Health Sciences, Ben-Gurion University, Beer-Sheva 84103, Israel. [3] Department of Molecular and Medical Pharmacology, David Geffen School of Medicine, University of California Los Angeles, Los Angeles, CA 90095, USA. [4] Department of Physiology and Cell Biology, Faculty of Health Sciences, Ben-Gurion University, Beer-Sheva 84105, Israel. [5] Institute of Medical Biochemistry Leopoldo de Meis, Universidade Federal do Rio de Janeiro, Rio de Janeiro, Brazil. ✉email: sekler@bgu.ac.il; Oshirihai@mednet.ucla.edu

The brown adipose tissue (BAT) dissipates energy in the form of heat in response to adrenergic stimuli triggered by cold exposure[1,2]. This pathway is initiated by the sympathetic neurotransmitter norepinephrine (NE), which induces a PKA-dependent lipid mobilization and oxidation. Resultant free-fatty acids (FFAs) activate BAT signature protein, uncoupling protein 1 (UCP1), leading to proton leak and uncoupled respiration[1,3,4]. In addition to their role in inducing UCP1 function, FFAs undergo β-oxidation and fuel the TCA cycle for production of reducing equivalents, to supply the electron transport chain (ETC), essentially utilized to drive uncoupled energy expenditure[2]. However, for the TCA cycle to keep up with the uniquely high energetic demand of thermogenesis, mitochondrial $Ca^{2+}$ is required for stimulating the activity of the dehydrogenases and ETC components[5–7]. Yet, uncontrolled or prolonged mitochondrial $Ca^{2+}$ rise, often encountered during ischemic[8] or neurodegenerative diseases[9], can lead to opening of the mitochondrial permeability transition pore (mPTP)[10,11].

The opening of mPTP has been associated with two hallmarks, the first is depolarization and a collapse in membrane potential, which is accompanied by excess of mitochondrial $Ca^{2+}$ accumulation[8,12]. The association between mitochondrial $Ca^{2+}$ overload and depolarization is not well understood, yet both stimuli were shown to be essential and sufficient for the pore opening, leading to mitochondrial swelling and triggering cytochrome $c$ release, thus impairing respiration and culminating in cellular death[12–14]. Thereby, the fine-tuning of mitochondrial $Ca^{2+}$ homeostasis in conjugation with thermogenesis has to be tight and well regulated to avoid detrimental effects on cell physiology.

Mitochondrial $Ca^{2+}$ uptake is mediated through a highly selective $Ca^{2+}$ channel, recently linked to the mitochondrial calcium uniporter (MCU) gene[15–17], $Ca^{2+}$ is subsequently transported out primarily by a mitochondrial $Na^+/Ca^{2+}$ exchanger termed NCLX[18]. These components were known for decades but their molecular identities have been discovered only recently[19].

The contribution of mitochondrial $Ca^{2+}$ elevation to cold-stimulated thermogenesis is poorly understood. Surprisingly, a recent study that explored mice lacking MCU in BAT found that $Ca^{2+}$ uptake through MCU is dispensable for BAT-mediated thermogenesis; neither a basal nor a cold-stimulated phenotype were found in these mice[20]. This observation, coupled with early mitochondrial $Ca^{2+}$ studies showing that $Na^+$-dependent mitochondrial $Ca^{2+}$ extrusion is essential for brown adipocyte (BA) uncoupled respiration[21], bring up the possibility that $Ca^{2+}$ extrusion rather than entry may be essential in BAT activation. Adrenergic stimulation of thermogenesis involves the activation of PKA, the induction of uncoupling, leading to mitochondrial membrane potential depolarization. A previous study showed that PKA-mediated phosphorylation of NCLX, at a serine residue (Ser258) located on the regulatory site of the protein rescued mitochondrial $Ca^{2+}$ efflux in depolarized neurons lacking PINK1[22]. However, the physiological role of this PKA-dependent regulation is still unclear.

In this study, we show that adrenergic signaling activates a NCLX-mediated mitochondrial $Ca^{2+}$ efflux required to establish both uncoupled energy expenditure and BAT cellular viability.

In the absence of adrenergic stimulation, deletion of the NCLX is inconsequential both in vitro and in vivo. While adrenergic stimulation of wild type (WT) BA elicits thermogenic response characterized by a robust increase in mitochondrial respiration, adrenergic stimulation of BA lacking NCLX results in $Ca^{2+}$ overload, mitochondrial swelling, and cytochrome $c$ release leading to cell death in BAT. However, despite the $Ca^{2+}$ overload, we found that mPTP inhibition fully restores the thermogenic capacity and maintains cellular viability of stimulated NCLX-null BA both in vitro and in vivo. Overall, this study reveals a key

pathway through which mitochondrial $Ca^{2+}$ efflux permits a robust activation of respiration in response to profound uncoupling, while preventing the activation of cell death pathways.

## Results

**Mitochondrial $Ca^{2+}$ extrusion is regulated by PKA activity in BAT.** While a rise in mitochondrial calcium has been documented during activation of thermogenesis in BAT, its functional role and regulation has not yet been elucidated. Recently Flicker et al.[20] found that mitochondrial $Ca^{2+}$ uptake through the mitochondrial calcium uniporter is inconsequential for BAT thermogenic function. However, the role of $Ca^{2+}$ extrusion pathway has not been investigated. To begin examining the role of $Ca^{2+}$ extrusion in BAT thermogenesis, we studied mitochondrial $Ca^{2+}$ in cultured BA during their response to NE. Noticeably, NE induces a prompt mitochondrial $Ca^{2+}$ uptake demonstrated by influx rate, faster than the subsequent $Ca^{2+}$ extrusion event, which is described by the efflux rate (Fig. 1a). This observation indicates that the $Ca^{2+}$ extrusion entity is the rate-limiting controller of mitochondrial $Ca^{2+}$ transients due to its slower activity. We first determined whether $Ca^{2+}$ efflux is $Na^+$ dependent. Intact BA were superfused with NE while monitoring mitochondrial $Ca^{2+}$ fluxes using the mitochondrial $Ca^{2+}$ dye, Rhod-2AM. We found that mitochondrial $Ca^{2+}$ release, but not its uptake, is dependent on $Na^+$ in BA. Substituting $Na^+$ iso-osmotically with $NMDG^+$ resulted in a strong reduction in mitochondrial $Ca^{2+}$ efflux. Therefore, we conclude that the mitochondrial $Na^+/Ca^{2+}$ exchanger is an essential mechanism for mitochondrial $Ca^{2+}$ extrusion in BA (Fig. 1b, c). Beyond inducing intracellular $Ca^{2+}$ rise, NE promotes an array of physiological changes in BA. To develop a cellular system where $Ca^{2+}$ rise can be induced without the activation of the entire thermogenic cascade, we utilized the purinergic agonist, ATP, which can induce intracellular $Ca^{2+}$ transients. ATP, a broadly studied purinergic agonist mobilizes $Ca^{2+}$ to the cytosol and subsequently to the mitochondria by depleting the ER stores via IP3R[23]. Unlike NE, ATP activity does not involve stimulation of Gs alpha subunit, mitochondrial depolarization, and induction of PKA signaling[23–25]. As expected, NE and ATP evoked a similar $Ca^{2+}$ uptake response. Surprisingly, however, as compared to NE, stimulation with ATP induced a significantly lower $Ca^{2+}$ efflux response, suggesting that NE mediates a pathway that activates $Ca^{2+}$ extrusion, which ATP cannot. We rationalized that the difference in the $Ca^{2+}$ extrusion pattern following stimulation with ATP as compared to NE is the result of bypassing the signal transduction pathway of PKA, which is physiologically activated by NE, but not by ATP. Indeed, pretreatment of cells with the PKA agonist, Forskolin (FSK), resulted in ATP and NE stimulation having an identical $Ca^{2+}$ extrusion pattern; supporting that under physiological activity of BAT, NE signaling is amplifying a PKA-mediated pathway that is required to upregulate mitochondrial $Ca^{2+}$ efflux (Fig. 1d–f). To further validate the role of PKA signaling in the activation of mitochondrial $Ca^{2+}$ efflux, we stimulated BA with NE in the presence or absence of the PKA inhibitor, H-89, and mitochondrial $Ca^{2+}$ was monitored. Mitochondrial $Ca^{2+}$ efflux, but not the influx, was entirely blocked by H-89, further supporting that PKA signaling is essential for regulating mitochondrial $Ca^{2+}$ efflux in BAT (Fig. 1g–i). Overall, these data show that NE regulates mitochondrial $Ca^{2+}$ efflux, but not influx, in a PKA-dependent manner.

**NCLX is essential for NE-induced uncoupled respiration in BA.** Next, we asked whether NCLX is essential for mitochondrial $Ca^{2+}$ efflux in activated BA and then tested its role in uncoupled respiration and energy expenditure. First, mitochondrial $Ca^{2+}$

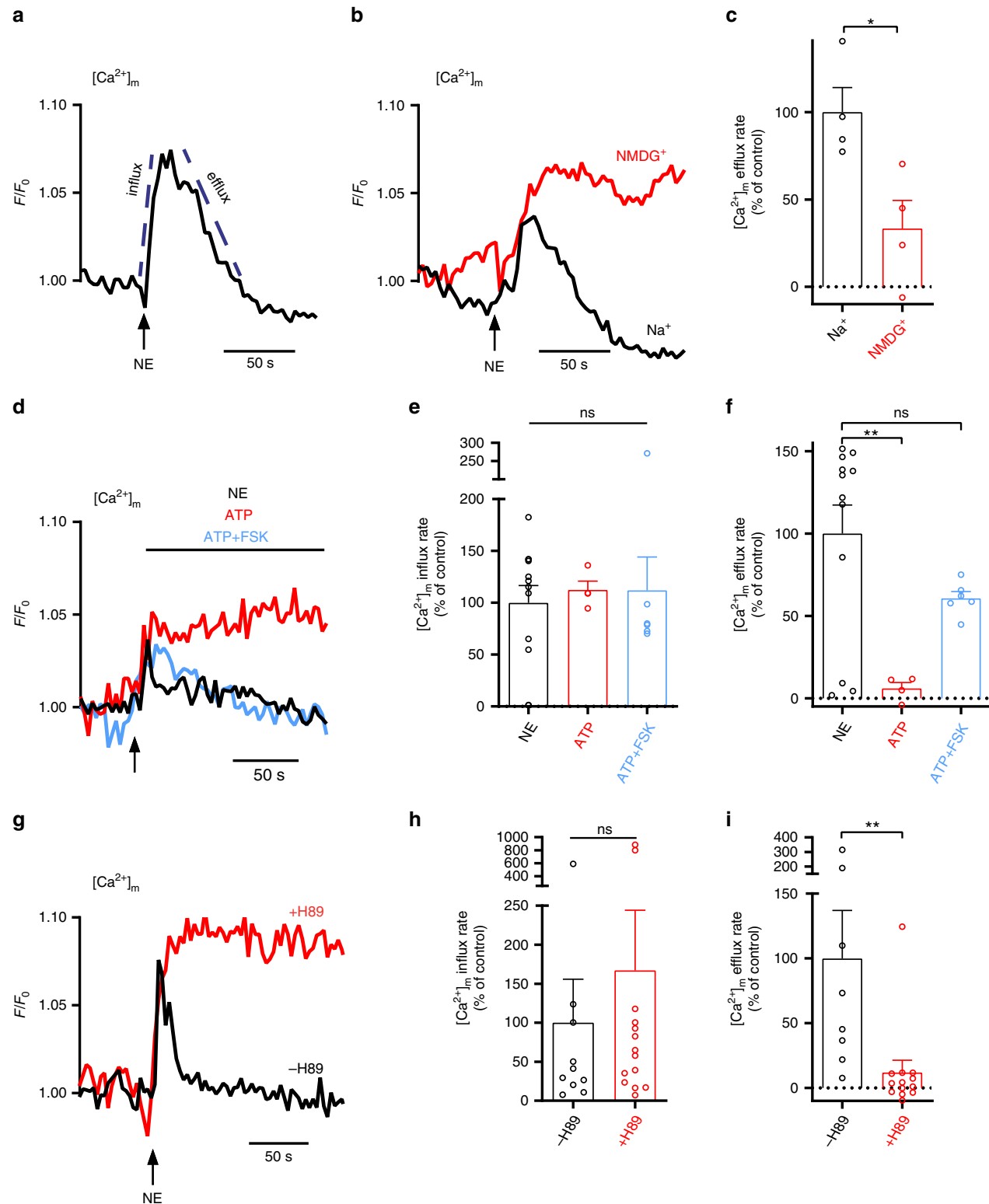

signaling was monitored in primary BA transfected with small interfering RNA, siNCLX, or siControl (Fig. 2a–c); and in isolated BA from Knock-Out (KO) and WT mice (Fig. 2d–i). In both experimental paradigms, NE-induced mitochondrial $Ca^{2+}$ efflux was totally lost when NCLX was deleted (Fig. 2c, i), resulting in a sustained mitochondrial $Ca^{2+}$ overload. Our results further show that adrenergic stimulation did not affect mitochondrial $Ca^{2+}$

uptake (Fig. 2b, h) or result in alterations of MCU expression levels tested in NCLX-ablated BA (Fig. 2e, f). These results were further verified in primary NCLX KO and WT primary BA transduced with the highly sensitive, genetically encoded mitochondrial $Ca^{2+}$ reporter GCaMP6-mt[26] (Supplementary Fig. 1a–d). Collectively, these data indicate that NCLX is the primary $Ca^{2+}$ exporter in BAT mitochondria.

**Fig. 1 Mitochondrial $Ca^{2+}$ extrusion in BA is mediated by $Na^+/Ca^{2+}$ exchange and requires PKA. a** Representative traces of mitochondrial $Ca^{2+}$ kinetics in primary brown adipocytes (BA) after application of NE (1.5 μM). Mitochondrial $Ca^{2+}$ was monitored using the mitochondrial $Ca^{2+}$ dye, Rhode2-AM. The dashed lines represent the linear fit used to calculate the $Ca^{2+}$ influx and efflux rates. **b** Representative traces of mitochondrial $Ca^{2+}$ transients upon application of NE to BA in the presence or absence of $Na^+$ ($NMDG^+$ iso-osmotically replacing $Na^+$). **c** Quantification of mitochondrial $Ca^{2+}$ efflux rates in the presence and absence of $Na^+$. $NMDG^+$ was used as a cationic replacement of $Na^+$ ($n = 4$ independent experiments per condition). **d** Representative traces of mitochondrial $Ca^{2+}$ transients upon application of NE (1.5 μM), ATP (100 μM), or ATP in cells pretreated with PKA activator, Forskolin (FSK (50 μM, 15 min)). **e, f** Quantification of mitochondrial $Ca^{2+}$ fluxes in response to NE, ATP, or ATP+FSK stimulation. NE ($n = 12$), ATP ($n = 4$), ATP+FSK ($n = 6$). **g** Representative traces of mitochondrial $Ca^{2+}$ transients in primary BA upon stimulation by NE in the presence and absence of PKA inhibitor, H-89 (5 μM, 1 h pre-incubation). **h, i** Quantification of mitochondrial $Ca^{2+}$ traces under treatment of H-89 ($n = 10$ and 14 for control and H-89 in **h**, $n = 8$ and 13 for control and H-89 in **i**). Student's $t$-test (**c, h, i**); one-way ANOVA with Tukey's post hoc test (**e, f**). Data are expressed as means ± SEM. $^{ns}p > 0.05$, $^*p < 0.05$, $^{**}p < 0.001$. Replicates are indicated by individual dots shown for each group in all graphs. Source data are available as a Source Data file.

Furthermore, monitoring cytosolic $Ca^{2+}$ transients after NE stimulation by using the Fura-2AM dye showed that while NE-dependent cytosolic $Ca^{2+}$ rise was similar in NCLX KO and WT BA, cytosolic $Ca^{2+}$ clearance rate was reduced in the NCLX KO BA compared to the WT BA (Supplementary Fig. 2a–c). The expression levels of major $Ca^{2+}$ regulators such as plasma membrane $Ca^{2+}$ ATPase (PMCA) or sarcoplasmic reticulum $Ca^{2+}$-ATPase 2 (SERCA2) and ryanodine receptors (RyRs) were unaltered in the NCLX KO BA compared to that of WT BA (Supplementary Fig. 2d–i). This finding is consistent with the previously described role of NCLX in controlling cytosolic $Ca^{2+}$ signaling observed in other cellular systems[27].

Subsequently, we investigated the bioenergetic consequences of NCLX deletion in primary BA. We hypothesized that NCLX activity will be required to maintain NE-stimulated mitochondrial uncoupling and energy expenditure in BA. To test this hypothesis, we compared NE-stimulated oxygen consumption rates (OCR) of primary BA from NCLX KO mice and WT mice. To further validate that NE-stimulated respiration was induced by uncoupling, we verified a negligible effect of ATP synthase inhibitor, oligomycin A, as shown in Fig. 2j. Our results show that deletion of NCLX resulted in 45% reduction in the respiratory response to NE (Fig. 2j, k). Suppressed respiratory response to NE could be a result of reduced respiratory chain capacity or due to decreased energy demand in the form of UCP1-mediated proton leak. To differentiate between the two possibilities, we induced uncoupled respiration using the pharmacological uncoupler, FCCP. Maximal capacity for uncoupled respiration was tested in BA by injection of FCCP into the respirometer chamber after NE. Our results show that adrenergically stimulated NCLX-null BA have a strong decrease in their mitochondrial maximal respiration compared to stimulated WT, with a maximum of 426.2 ± 36.86% of basal respiration for WT compared to 173.8 ± 19.37% for NCLX-null BA (Fig. 2j, l). We further interrogated the role of $Na^+/Ca^{2+}$ exchange in primary BA by subjecting WT cells to a $Na^+$-containing medium or a $Na^+$-depleted medium; the latter was achieved by iso-osmotically substituting $Na^+$ with $NMDG^+$. Similar to results obtained in cells lacking NCLX, we observed that removal of $Na^+$ in WT BA lead to a reduction in adrenergic respiratory response, followed by suppression of maximal respiration (Supplementary Fig. 3a–d). Unlike the NCLX-null BA, however, basal respiration was reduced in this experimental system, which can be attributed to NCLX-independent effects of lack of $Na^+$ on metabolic functions (Supplementary Fig. 3a). In all, we show that NCLX is essential for BA response to NE, preventing mitochondrial $Ca^{2+}$ overload and enabling the induction of energy expenditure.

**NCLX is essential for non-shivering thermogenesis in vivo.** Based on the observed impairment in adrenergic-stimulated uncoupled respiration in NCLX-null BA in vitro, we assessed the role of NCLX in BAT thermogenesis by comparing NCLX KO mice to their WT. We used several complementary approaches to evaluate BAT energy expenditure and thermogenic capacity in mice with whole-body deletion of NCLX. First, we determined defense of body temperature after acute response to cold stress by continuously measuring mice core body temperature at ambient 4 °C. Although both genotypes had similar core body temperature at time zero (Supplementary Fig. 4a), our results show impaired thermoregulation in NCLX KO male mice after acute cold exposure for 6–8 h, where 67% of the mice had to be pulled out of the cold chamber due to their failure to maintain body temperature. This is depicted in the survival curve where the term survival indicates the capacity to maintain body temperature above 28 °C (Fig. 3a, b). This finding was further confirmed in another cohort of NCLX KO and WT male littermates (Supplementary Fig. 5a, b). We then monitored mice whole-body oxygen consumption during cold stress at 4 °C. No difference in the mice oxygen consumption was observed at the beginning of the cold exposure. However, after 7 h of cold exposure, whole-body oxygen consumption was significantly reduced in NCLX KO compared to WT mice, with substantial loss of more than 40% of their $O_2$ consumption (Fig. 3c).

To determine the isolated effect of NCLX deletion on non-shivering thermogenesis, we utilized the selective β3-adrenergic agonist, CL-326,243, allowing for a direct measurement of non-shivering thermogenesis. CL-326,243 was injected in anesthetized mice kept at thermoneutral 30 °C to prevent neuronal stimulation of shivering and non-shivering thermogenesis, thereby eliminating the basal activity of muscles or BAT. Consistent with our cold-stress experiments, we observed in two different cohorts, a significant lower $O_2$ consumption in NCLX KO mice after β-3 stimulation as compared to their WT mice, pointing to a non-shivering thermogenesis dysfunction in these mice (Fig. 3d–f and Supplementary Fig. 5c–e).

To determine the gene-dose-dependent effect we proceeded with the evaluation of the heterozygous deletion of NCLX (NCLX +/−). Thermogenic capacity of NCLX+/− mice was also reduced in comparison to their WT littermates, exhibiting significantly lower survival rates and core body temperatures after 6–8 h of cold stress compared to their WT littermates (Fig. 3g, h).

Furthermore, BAT-specific adrenergic activation induced by CL-326,243 injection resulted in lower oxygen consumption rates in NCLX+/− mice as compared to WT (3232.5 ± 332 and 2536.7 ± 181 ml/kg/h for WT and NCLX+/−, respectively). These results further support the hypothesis that the NCLX activity is indispensable for non-shivering thermogenic function (Fig. 3i, j).

Moreover, to determine if the phenotype of NCLX KO mice is gender-dependent, we tested our in vivo results in female littermates of WT, NCLX+/−, and NCLX KO. We found that both NCLX+/− and NCLX KO female mice are cold-intolerant and have a worse survival rate as compared to their WT littermates (Supplementary Fig. 6a, b).

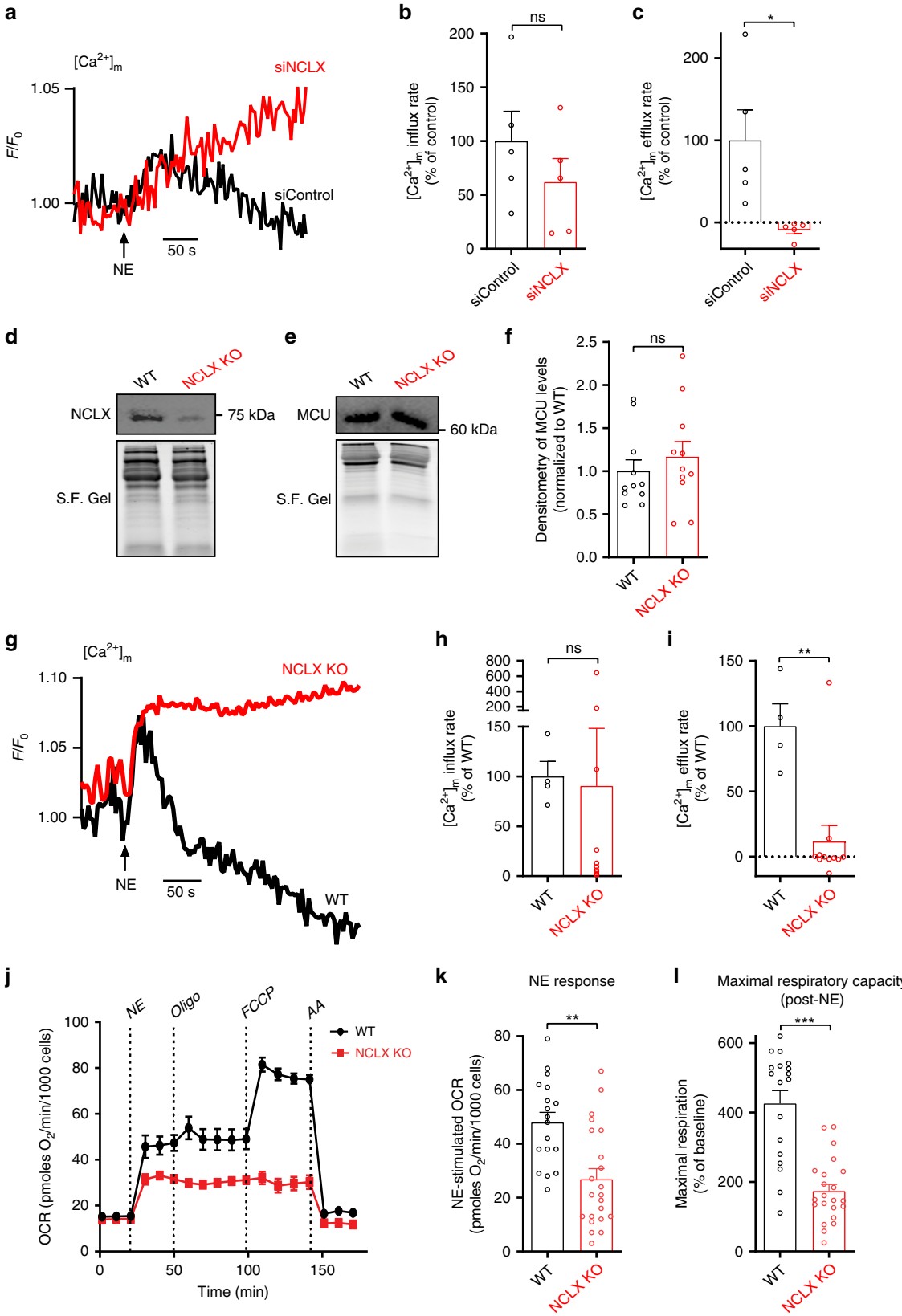

Overall, these results indicate that mitochondrial Na$^+$/Ca$^{2+}$ exchange is essential for adrenergic uncoupled respiration and activation of non-shivering thermogenesis.

**Bioenergetic defects in NCLX-null BA appear following adrenergic stimulation**. To investigate the mechanism underlying the suppression of NE-induced uncoupled respiration triggered by

**Fig. 2 NCLX is required for mitochondrial uncoupled respiration in primary BA. a** Representative traces of mitochondrial $Ca^{2+}$ fluxes in primary BA transfected either with siNCLX or siControl and monitored for mitochondrial $Ca^{2+}$ transients evoked by NE. **b, c** Quantification of influx and efflux rates in primary BA transfected either with siNCLX or with siControl ($n = 5$ independent experiments per condition). **d** Western blot analysis of NCLX levels from NCLX KO and WT BAT. Stain-Free Gels (S.F. Gels) were used as a normalization method for total loaded protein. **e, f** Western blot analysis of MCU and quantification of its relative levels ($n = 11$ per group). **g** Representative mitochondrial $Ca^{2+}$ transients of primary BA from NCLX KO and WT mice induced by NE. **h, i** Quantification of $Ca^{2+}$ influx and efflux rates in NCLX KO and WT BA. Mitochondrial $Ca^{2+}$ influx rates were unaffected while $Ca^{2+}$ efflux rates were diminished in NCLX KO BA ($n = 4$ for WT and $n = 11$ for NCLX KO). **j** Representative traces of oxygen consumption rates (OCR) of cultured BA from NCLX KO and WT mice. Stimulation by NE was the first injection. OLIGOmycin was used to assess mitochondrial uncoupling efficiency. FCCP was then injected to assess maximal respiration and non-mitochondrial OCR was evaluated by Antimycin A injection (AA). **k, l** Quantification of NE response and maximal respiration after NE stimulation ($N = 3$ independent experiments with $n = 18$ for WT and $n = 22$ for NCLX KO). Student's $t$-test (**b, c, f, h, i, k, l**); data are expressed as means ± SEM. $^{ns}p > 0.05$, $^{*}p < 0.05$, $^{**}p < 0.001$, $^{***}p < 0.0001$. Replicates are indicated by individual dots shown for each group in all graphs. Source data are available as a Source Data file.

the deletion of NCLX, we first looked for changes in UCP1 expression levels. However, we were unable to detect significant changes in protein expression levels in NCLX KO BA as compared to WT BA (Fig. 4a, b). Since a reduction in mitochondrial mass can contribute to the loss of respiratory capacity, we assessed mitochondrial mass by two independent approaches: (1) western blot analysis of the mitochondrial protein importer TOM20 and (2) staining with Mitotracker Green (MTG), a fluorescent dye that covalently binds to the mitochondria independently of its membrane potential. These experiments showed no indicative changes in mitochondrial mass (Fig. 4c–e).

Moreover, to investigate basal differences in mitochondrial membrane potential in NCLX KO and WT BA, cells were stained with the membrane potential-sensitive dye TMRE. To control for focal plane and analyze morphometric details, cells were co-stained with MTG[28]. Deletion of NCLX did not affect mitochondrial membrane potential as revealed by fluorescence microscopy of TMRE-to-MTG ratio. However, we found minor changes in mitochondrial architecture, including a decrease of 0.93% in the parameter of aspect ratio and an increase of 1.033% in mitochondrial roundness in the NCLX KO BA indicative of a subtle phenotype at resting state (Fig. 4f–j).

Furthermore, expression levels of OXPHOS complex subunits and assembly of super complexes I+III before and after adrenergic stimulation of BA from NCLX KO and WT were unchanged (Fig. 4k–n).

Assessment of mitochondrial complex I, II, and IV activity revealed a significant reduction in complex I activity in non-stimulated NCLX KO mitochondria compared to WT (Fig. 4o).

Interestingly, however, basal and spare respiratory capacity before NE stimulation were unaffected in intact NCLX KO BA compared to WT BA. Yet, NE stimulation of NCLX KO led acutely to a collapse of bioenergetics and disability to meet the energy demand (Fig. 4p, q, Supplementary Fig. 7a).

Overall, these results indicate that the mechanism underlying the bioenergetics defect triggered by NCLX deletion is acutely induced by NE and does not involve changes to UCP1 and OXHPOS expression levels or mitochondrial mass.

**NCLX is essential for preserving BAT cell viability during thermogenesis.** Mitochondrial $Ca^{2+}$ overload can impair respiratory capacity by the induction of permeability transition and mPTP opening. Activation of mPTP involves mitochondrial swelling, release of cytochrome $c$ and can eventually lead to cellular death[8,14,29–31]. We hypothesized that $Ca^{2+}$ overload, induced by NE stimulation in NCLX-null BA, may induce permeability transition leading to suppression of thermogenesis.

To test this hypothesis, we employed complementary in vitro and in vivo models. First, we imaged NCLX KO and WT BA before and after 6 h of NE stimulation. Cells were stained with

TOM20 to mark mitochondrial network at the outer membranes and for cytochrome $c$.

Super-resolution confocal microscopy of BA revealed a NE-induced release of cytochrome $c$ from the mitochondria of the NCLX-null cells but not the WT controls (Fig. 5a, b). In addition, loss of mitochondrial cytochrome $c$ was accompanied by a dramatic mitochondrial swelling and mitochondrial rupture of the outer membrane, occurring only under adrenergic stimulation of NCLX-null cells (Fig. 5a, c, Supplementary Fig. 8a, b, Supplementary Data 1). To interrogate mitochondrial swelling at high temporal and spatial resolution in adrenergically stimulated NCLX-null BA, mitochondrial outer membranes were labeled with mCherry-Fis1 probe. Time-lapsed imaging showed that mitochondrial swelling of NCLX KO occurs in a spatially organized manner that starts at one pole and gradually propagates to the center of the cell (Fig. 5d). To confirm our super-resolution microscopy observation of mitochondrial swelling in NCLX KO BA, we carried out electron microscopy analysis for NE-stimulated WT and NCLX KO BA. While mitochondrial morphology in WT BA was retained, mitochondrial cristae density in NCLX-null BA was reduced by 39% (Fig. 5e, f). The disruption of cristae was accompanied by a significant increase of mitochondrial swelling and rupture (Fig. 5e–h).

Collectively, these results strongly suggest that stimulated BA require NCLX activity to prevent mitochondrial swelling and loss of cytochrome $c$.

To further investigate the association between NE-mediated swelling in NCLX KO BA and $Ca^{2+}$ overload, we applied different $Ca^{2+}$ chelators that bind to $Ca^{2+}$ at the different cellular levels (Fig. 6a); EGTA buffers extracellular $Ca^{2+}$, BAPTA-AM buffers intracellular $Ca^{2+}$, and finally we inhibited $Ca^{2+}$ entry to the mitochondria using an analog of ruthenium red, Ru360. NE-stimulated NCLX KO BA pretreated with vehicle had a swelling rate of 61%. However, extra or intracellular $Ca^{2+}$ chelation, as well as the blockage of mitochondrial $Ca^{2+}$ entry, prevented mitochondrial swelling in NE-stimulated NCLX KO BA (Fig. 6b, c).

The observation of mitochondrial swelling and cytochrome $c$ release in NE-stimulated NCLX-null cells in culture suggested that cold exposure may lead to cellular death in BAT. To determine the effect of adrenergic stimulation on cell viability in vivo we applied an intermittent cold exposure protocol with the goal of allowing prolonged activation of BAT while preventing hypothermic mortality. Animals were cold stressed at 4 °C for 5 days with recovery periods after a 4–6 h of cold-stress till they gained back their core body temperature. On day 5, BAT was harvested and cellular death was assessed by TUNEL staining of BAT. Evaluation of positive cells per mm² (Fig. 6d, e) and the percentage of TUNEL+ cells (Fig. 6d, f) demonstrate a remarkable cellular death rate of BA from

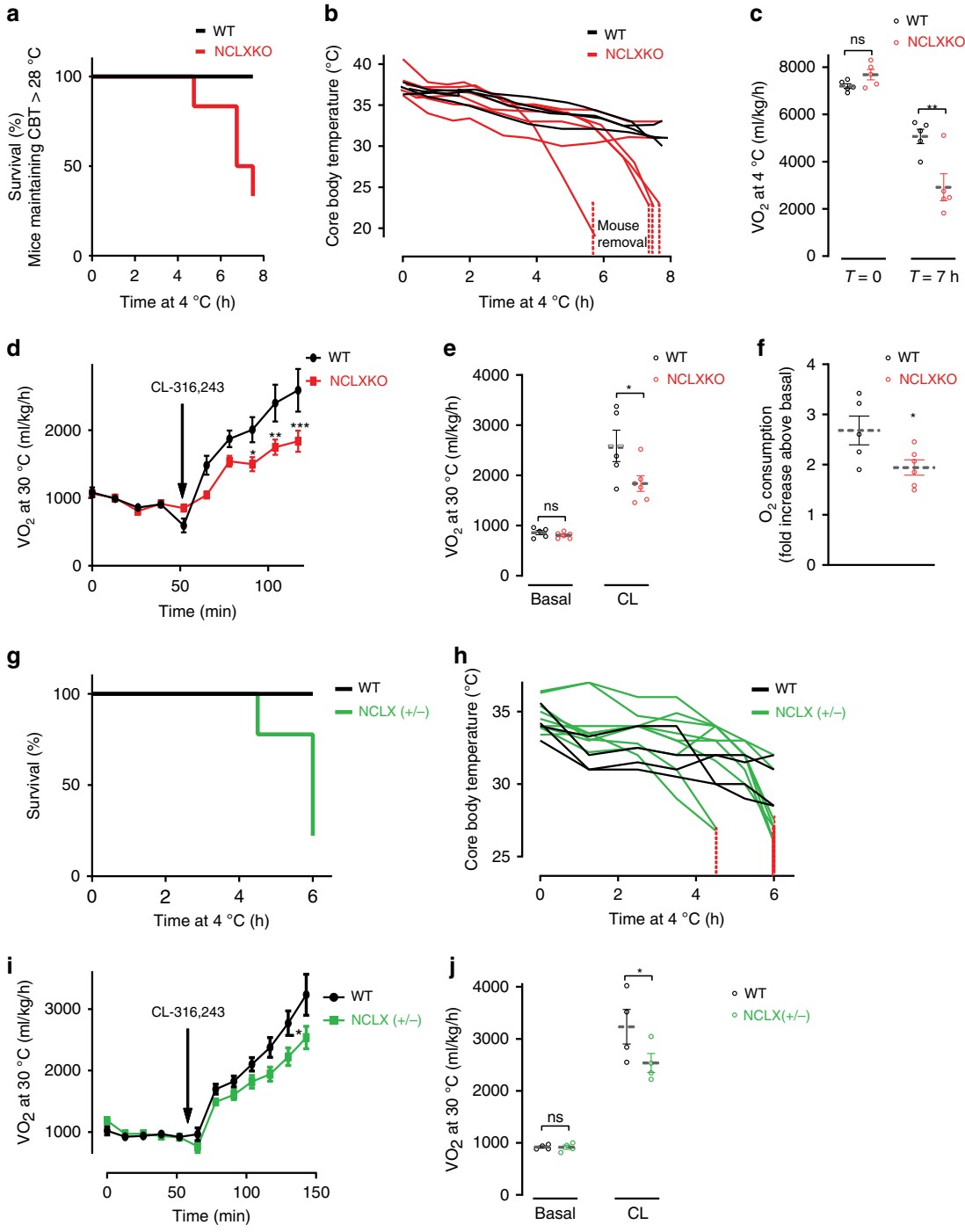

**Fig. 3 Thermogenesis is impaired in NCLX-null mice. a** Survival curves of 8–10-week-old NCLX KO and WT male mice cold stressed at 4 °C. Animals reaching temperatures 28 °C and below were returned to room temperature for recovery and counted as drop-outs. Survival rates depict the rate at which animals had to be removed from 4 °C to room temperature to avoid mortality ($n = 4$ mice for WT and $n = 6$ mice for NCLX KO). CBT, core body temperature. **b** Core body temperature traces of the animals from experiment (**a**) during cold exposure (4 °C). The dashed red line indicates animal removal. **c** VO$_2$ of NCLX KO and WT mice subjected to cold (4 °C) at $t = 0$ and $t = 7$ h ($n = 5$ mice per group). **d**–**f** Non-shivering thermogenesis of NCLX KO and WT mice assessed by measuring O$_2$ consumption at baseline and followed by β-3 adrenergic stimulation (CL-316,243 injection (1 mg/kg)) in anesthetized mice at 30 °C. **d** Traces of VO$_2$ of NCLX KO and WT mice. **e** VO$_2$ at baseline and under CL stimulation. **f** Fold increase of O$_2$ consumption after the CL stimulation ($n = 5$ mice for WT and $n = 6$ mice for NCLX KO). **g** Survival curves of WT and NCLX heterozygous ($+/-$) littermates cold stressed at 4 °C; mice reaching 28 °C or lower were returned to room temperature for recovery and counted as drop-outs ($n = 4$ mice for WT and $n = 9$ mice for NCLX+/−). **h** Core body temperature of the animals shown in **g**. **i** VO$_2$ of WT and NCLX+/− mice at basal and after CL-316,243 injection (1 mg/kg), under anesthesia at 30 °C. Traces of VO$_2$ of NCLX+/− and WT mice. **j** VO$_2$ at baseline and under CL stimulation ($n = 4$ mice per group). Student's $t$-test (**c**, **e**, **f**, **j**); two-way ANOVA (**d**, **i**). Data are expressed as means ± SEM. $^{ns}p > 0.05$, $^*p < 0.05$, $^{**}p < 0.001$, $^{***}p < 0.0001$. Replicates are indicated by individual dots shown for each group in all graphs. Source data are available as a Source Data file.

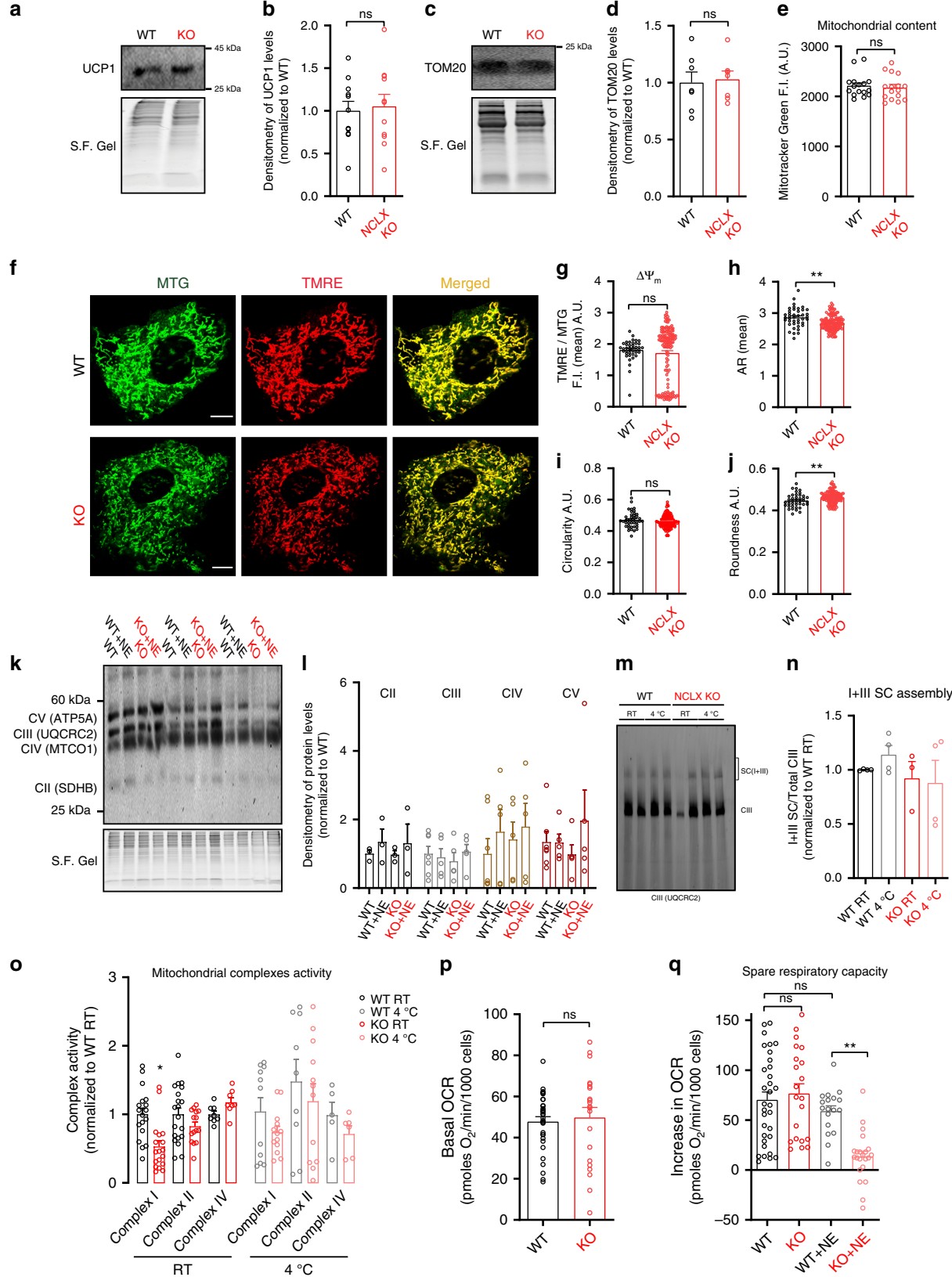

cold-stressed NCLX KO mice as compared to WT (Fig. 6d, Supplementary Fig. 9a). Importantly, minimal and non-significant cellular death rates were observed in NCLX KO BAT from mice housed at room temperature of 25 °C or thermoneutrality (30 °C) compared to WT BAT from mice

housed at these conditions, indicating that cellular death ensues only when BAT is activated (Supplementary Fig. 9b–g).

Overall, these results indicate that NCLX activity, in vitro and in vivo, is critical for BA survival during adrenergic stimulation by preventing the ensuing mitochondrial $Ca^{2+}$ overload.

**Fig. 4 Respiratory dysfunction mediated by NCLX loss in BA is acutely induced following adrenergic stimulation. a, b** Western blot analysis of NCLX-null and WT BA for UCP1 expression levels S.F. Gel was used as a normalization reference for total protein loading ($n = 11$ per group). **c, d** Western blot analysis of NCLX-null and WT BA for TOM20 expression levels ($n = 7$ per genotype). **e** Mitotracker Green (MTG) fluorescence intensity in NCLX-null and WT primary BA, measured by Operetta high-content imaging system ($n = 16$ per condition from total $N = 3$ imaging experiments). **f–j** Membrane potential and morphometric analyses of mitochondrial morphology in NCLX-null and WT BA ($N = 4$ independent experiments and $n =$ at least 41 cells per condition). **f** Representative images of NCLX-null and WT BA stained with TMRE and MTG. Scale bar, 10 μm. **g** Quantification of mitochondrial membrane potential of NCLX-null and WT BA. TMRE signal is normalized to MTG signal to generate a ratio that avoids focal plane artefacts. **h** Quantification of mitochondrial aspect ratio (AR), **i** circularity and **j** roundness in NCLX-null and WT BA using the MTG channel. **k, l** Western blot analysis of primary NCLX-null and WT BA for OXPHOS complex subunits II–V (CII–CV). Cells were treated with or without NE for 6 h ($n = 3-7$ per group). **m** Western blot of super complex (SC) I+III in BAT homogenates from cold-exposed and non-exposed NCLX-null and WT mice. **n** Quantification of the fraction of complex III that is assembled into SC(I+III) relative to total complex III from cold-exposed and non-exposed NCLX-null and WT mice ($N = 3-4$ mice per condition). **o** Assessment of mitochondrial complexes activity in cold-exposed and non-exposed NCLX-null and WT BAT: CI ($n = 11-18$), CII ($n = 9-20$), and CIV ($n = 5-8$). **p** Quantification of basal respiration in intact non-stimulated BA ($N = 6-8$ independent experiments with $n = 32$ for WT and $n = 21$ for NCLX KO). **q** Quantification of maximal spare respiratory capacity in NE-stimulated and non-stimulated NCLX-null and WT BA. Maximal respiration was induced by FCCP ($N = 3-8$ independent experiments with $n = 18-32$ per condition). Student's *t*-test (**b, d, e, g, h, i, j, p**); one-way ANOVA with Tukey's post hoc test (**l, n, o, q**). Data are expressed as means ± SEM. $^{ns}p > 0.05$, $^{**}p < 0.001$. Replicates are indicated by individual dots shown for each group in all graphs. Source data are available as a Source Data file.

**Blockage of mPTP rescues uncoupled energy expenditure in NCLX-null BAT**. The findings described in Fig. 5 showing mitochondrial swelling and cytochrome *c* release in activated NCLX-null BA suggest that in the absence of NCLX, stimulation with NE leads to mitochondrial $Ca^{2+}$ overload and the induction of permeability transition. A key regulator of the mPTP opening is Cyclophilin D (CypD)[32–34]. Inhibitors of CypD such as Cyclosporin A have off-targets by interfering with Cyclophilin A and with Calcineurin[35]. However, a novel and specific Cyclosporine derivative, named NIM811, has been shown to effectively block $Ca^{2+}$-induced mPTP opening, without exerting inhibitory side-effects on Calcineurin[35,36].

To test the role of mPTP in mediating the bioenergetics phenotype of NCLX-null BA, we tested the effect of NIM811 on BA response to NE. Primary NCLX KO and WT BA were pretreated either with vehicle (DMSO) or NIM811 and subjected to OCR measurements. Remarkably, NIM811 pretreatment fully restored and rescued NE response in NCLX KO BA with rates similar to the WT BA (Fig. 7a, b). Moreover, this effect was followed by a recovery of maximal respiration (Fig. 7a). Respiratory rates during NE stimulation were not affected by NIM811 in BA of WT mice (Fig. 7b, Supplementary Fig. 10a).

Surprisingly, assessment of mitochondrial $Ca^{2+}$ in NCLX KO BA pretreated with NIM811 showed a sustained $Ca^{2+}$ overload to similar levels of vehicle-pretreated NCLX KO BA (Fig. 7c, d). Interestingly, however, when we assessed mitochondrial swelling after 4–6 h of NE treatment, NIM811 was sufficient to rescue NCLX KO BA (Fig. 7e, f).

Furthermore, NIM811 application reduced the mortality rate of NCLX KO BA to that of WT, assessed by in vitro TUNEL assay or mitochondrial staining, 72 h after NE stimulation (Fig. 7g, h).

We then asked whether NIM811 treatment can rescue BAT function of NCLX-null mice in vivo. We pretreated NCLX KO mice either with vehicle or NIM811 ($50 \, mg \, kg^{-1}$) and compared their thermogenic function to WT mice, this was followed by the measurement of BAT glucose uptake during cold exposure, assessed by μPET/CT imaging with $^{18}F$-fluorodeoxyglucose ($^{18}F$-FDG) analog (Fig. 7i). While cold-stress of 6–8 h exhibited a worse survival rate and impaired cold tolerance of NCLX KO treated with a vehicle, NIM811-treated NCLX KO mice displayed an increased overall survival rate, with values remarkably similar to WT animals (Fig. 7j, Supplementary Fig. 11a).

We next determined cold-induced BAT activity, assessed by in vivo glucose uptake utilizing the glucose tracer, $^{18}F$-FDG. While NCLX KO mice showed a reduction of 60.33% ± 18.67% in BAT glucose uptake as compared to WT, NIM811 pretreated

NCLX KO mice almost fully recovered their BAT glucose uptake reaching values similar to those found in WT mice (Fig. 7k, l, Supplementary Movies 1–3).

Overall, these data suggest that blockage of the mPTP by NIM811 can restore NE-induced mitochondrial respiration in NCLX KO BA and rescue thermogenesis in both in vitro and in vivo systems, thus confirming the role of the mPTP in the pathogenic effect of NCLX deficiency.

## Discussion

Activation of thermogenesis involves cytosolic and mitochondrial calcium elevations, supporting a surge in oxygen consumption and energy expenditure, induced by adrenergic stimulation of mitochondrial uncoupling, fragmentation, and depolarization[5,6,37–40]. Yet, similar transitions involving uncoupled mitochondrial respiration, fragmentation, and depolarization are found in other cells to accompany apoptosis mediated by permeability transition[8–10]. The mechanism by which brown adipocytes avoid death while responding to a thermogenic signal has not been elucidated. In this study, we have identified a mitochondrial $Ca^{2+}$ extrusion pathway that is activated hormonally by the neurotransmitter NE in BA, and is mediated by the mitochondrial $Na^+/Ca^{2+}$ exchanger, NCLX. Identification of NCLX as the molecular entity regulating mitochondrial $Ca^{2+}$ efflux during thermogenesis allowed us for the first time to look into the mechanism by which BA regulate mitochondrial $Ca^{2+}$ elevation and prevent depolarization-induced permeability transition and death during activation of thermogenesis (Fig. 8, Supplementary Fig. 12).

In vitro and in vivo experiments in this study show that stimulation of $Ca^{2+}$ extrusion is indispensable for physiological uncoupled energy expenditure in BA. At first, the results presented here might contrast with a recent report showing in vivo that BAT-MCU KO mice are as cold tolerant as their WT littermates and presenting no basal or adrenergic-stimulated phenotype[20]. However, while BAT function seems to be independent of mitochondrial $Ca^{2+}$ uptake by MCU, once it is present in the mitochondrial matrix, $Ca^{2+}$ levels must be tightly regulated by NCLX activity to prevent $Ca^{2+}$ overload, activation of mPTP and ultimately cell death. Mechanistically, our study is the first study that shows a physiological role of PKA-regulated mitochondrial $Ca^{2+}$ extrusion via NCLX activation in BAT; this is in agreement with a previous report where NCLX was shown to be activated artificially by PKA activators in a Parkinson model[22]. Furthermore, we find that adrenergic stimulation of NCLX-null BAT impairs mitochondrial

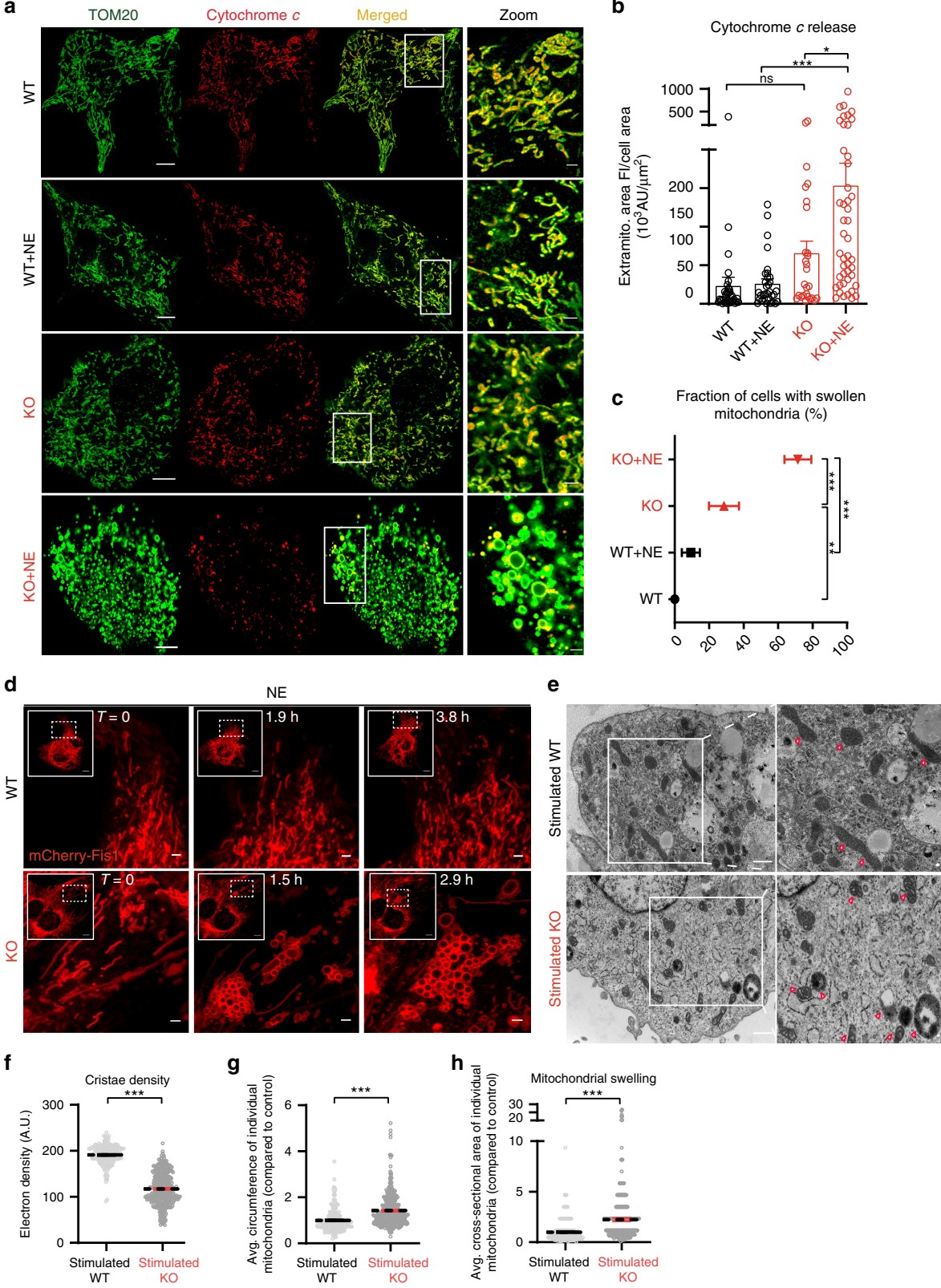

bioenergetics through the activation of permeability transition, resulting in mitochondrial swelling and cytochrome *c* release, followed by cell death. Our study and a recent work in a cardiac-induced model of NCLX KO[41] suggest that NCLX acts as an anti-apoptotic mechanism by inhibiting mitochondrial Ca$^{2+}$ overload. Moreover, our study provides the first evidence

of a hormonal-stimulus resulting in mitochondrial Ca$^{2+}$ overload leading to cell death. Our result, consistent with previous studies demonstrating that, apart from its stimulatory thermogenic role, NE functions as a pro-survival hormone for BAT[2,42]. Here we demonstrate that NE controls cell fate through the regulation of mitochondrial Ca$^{2+}$ handling.

**Fig. 5 Adrenergic stimulation in NCLX-null BA leads to mitochondrial swelling and cytochrome *c* release. a–c** Super-resolution confocal images of fixed WT or NCLX-null BA, with or without adrenergic stimulation for 6 h. **a** NCLX KO and WT BA treated with or without NE and immunostained for cytochrome *c* (red). Samples were co-stained for TOM20 to label mitochondrial localization. Scale bar is 10 μm, for the zoom images scale bar is 2 μm. **b** Quantification of cytochrome *c* release. Cytochrome *c* in the cytosol was calculated by measuring its total integrated fluorescence intensity (FI) outside the mitochondria per cell, normalized to total cellular area ($n = 27$–46 cells for each of the four conditions). **c** Quantification of mitochondrial swelling. Zero indicates bars where quantification yielded zero events of swelling ($n = 28$–37 cells for each of the four conditions). **d** Time-lapse of swelling formation following NE stimulation in NCLX-null and WT BA. Cells were labeled with mCherry-Fis1 probe marking their outer mitochondrial membrane. Swelling spatially starts at one pole of the cell and gradually propagated to the center of the cell. Scale bar, 10 and 2 μm for the zoom images. **e–h** Electron microscopy of NE-stimulated NCLX-null and WT BA. **e** Representative images of NE-stimulated NCLX-null and WT mitochondria of primary BA. Note that while the WT mitochondria are intact, adrenergic stimulation leads to cristae disruption, mitochondrial swelling, and rupture in the NCLX KO mitochondria. Scale bar, 1 and 0.1 μm for the zoom images. **f** Quantification of cristae density. **g** Circumference and **h** mitochondrial swelling assessed by cross-sectional area of individual mitochondria ($n = 185$ mitochondria at least were analyzed per condition). Student's *t*-test (**f, g, h**); one-way ANOVA with Tukey's post hoc test (**b, c**). Data are expressed as means ± SEM. $^{ns}p > 0.05$, $^*p < 0.05$, $^{**}p < 0.01$, $^{***}p < 0.0001$. Replicates are indicated by individual dots shown for each group in all graphs. Source data are available as a Source Data file.

Remarkably, this study shows that mitochondrial Ca$^{2+}$ overload is not deleterious per se for mitochondrial bioenergetics as long as the mPTP remains closed. Pharmacological inhibition of the mPTP by NIM811 was able to completely rescue the adrenergically mediated bioenergetics defects and the thermogenic dysfunction of NCLX KO BAT, in in vitro as well as in vivo. This is in agreement with a recent study showing that NIM811 can prevent cell death induced by mitochondrial Ca$^{2+}$ overload in the liver and mediated by the loss of MCU regulator, Micu1[36].

Our use of a mouse model where NCLX is deleted in all tissues raises the possibility that the observed impaired non-shivering thermogenesis found in these mice may not be a BAT-selective effect, but rather due to the involvement of other vital tissues. This includes a potential effect of beige fat, as beiging was recently linked to Ca$^{2+}$ cycling[40]. We present several experiments to support that the impaired thermogenesis in vivo involves a major BAT dysfunction component. First, non-shivering thermogenesis assessment by β-3 stimulation and $^{18}$F-FDG uptake in BAT, are organ-specific measurements in BAT. Second, our in vitro system of isolated NCLX KO BA, demonstrating an impaired energy expenditure and remarkable mitochondrial damage, recapitulate the in vivo phenotype.

Furthermore, while acute cardiac loss of NCLX resulted in death, it was shown that early embryonic deletion of NCLX in the heart resulted in complete viability, indicating that undetermined compensatory mechanisms are taking place in vital tissues of mice born with the deletion of NCLX[41]. However, primary BA isolated from NCLX KO mice show a complete loss of mitochondrial Ca$^{2+}$ efflux without interference in its uptake, ruling out such compensatory effect in BAT.

Lastly, BAT was shown to have a pivotal role in controlling circulating glucose levels, insulin sensitivity, and cardiometabolic health[43]. Therefore, to determine the role of NCLX in obesity and BAT long-term recruitment, future studies have to be done in an organ-specific KO model to control for the potential contribution of other organs such as the white and beige adipose tissues or the skeletal muscle.

In summary, the results of this study reveal that NCLX plays an essential role in regulating thermogenic BAT metabolic function and viability. The physiological importance of PKA-regulation of NCLX and mPTP inhibition shown in the study open future avenues for targeted therapeutic strategies to overcome pathological conditions in which impaired metabolic status is triggered by perturbed mitochondrial Ca$^{2+}$ homeostasis.

## Methods

**Experimental animals**. Experimental procedures conducted on mice were performed in accordance with animal welfare and in compliance with other related ethical regulations. The mice studies were conducted under an approved

Institutional Animal Care and Use Committee (IACUC) protocol at the University of California, Los Angeles (UCLA) and Ben-Gurion University. The mice were congenic to the C57BL/6NJ background, fed standard chow diet, and maintained under controlled conditions (housing at 22 °C with a 12:12 h light:dark cycle). In all experiments, mice were age and gender matched. For in vivo experiments, age-matched male and female mice of 10–12 weeks old were used.

WT C57BL/6NJ mice and NCLX-null (C57BL/6NJ-*Slc8b1*$^{em1(IMPC)}$/J) were purchased from Jackson laboratories (Jackson lab, Bar Harbor, ME), and bred in our vivarium. Mice genotyping was performed on earpieces or clipped tails obtained during the weaning of pups. Genotyping was performed following the protocol of Jax laboratories by real-time polymerase chain reaction, using a commercial vendor (Transnetyx). The following primers were used to detect NCLX-null−/−, Heterozygous+/−, or WT+/+ mice:

Forward primer-GGCTCCTGTCTTCCTCTGTG and Reverse primer-GTGTC CATGGGCTTTTGTG.

For in vivo experiments and analysis a randomization and a double blinded-manner were performed using ear-tagging and random mice numbering systems which were revealed after the termination of the experiment.

**NIM811 drug preparation and delivery**. For mPTP studies, mice were injected subcutaneously above the dorsal brown fat at a 50 mg/kg final dose of NIM811 (Novartis). The injection was made once a day for 5 days before cold-stress or μPET/CT experiments. The drug was diluted in a solution containing sterile saline, autoclaved Cremophor EL (15%, v/v) (Kolliphor EL, Sigma, C5135) and sterile ethanol (5%, v/v) to facilitate administration of NIM811 and its suspension stability.

**Thermogenesis and acute cold exposure**. Cold exposure experiments were performed as described previously[44]. In brief, subcutaneous, biocompatible, and sterile microchip transponders (IPTT-300; Bio Medic Data Systems, Seaford, DE, USA) were implanted in male and female mice in all groups at least 5 days prior to experimentation. On the day of the experiment, mice were housed singly in pre-chilled cages at 4 °C with free access to water. Body temperature was assessed every 30–45 min for 6–8 h using a wireless reader system (DAS-8007; Bio Medic Data Systems).

According to our IACUC protocol we defined "survival" when mouse body temperature remained >28 °C. At any temperature equal or below 28 °C, mice were rescued by removal to room temperature. The time of removal was recorded and considered as drop-out.

For experiments in which cell viability was detected in vivo, we applied an intermittent cold exposure which allows for longer-term cold exposure while preventing animal death. Animals were exposed to cold at intervals of 4–6 h intermitted by recovery periods till the mice gained back their core body temperature on a course of 5 days.

**In vivo measurement of BAT function**. In order to evaluate non-shivering thermogenesis and BAT function, we measured whole-body O$_2$ consumption at basal and in response to β3-adrenergic agonist, CL-316-243 (Sigma, C5976), in anesthetized mice as described previously[45]. Briefly, mice were housed at thermoneutrality (30 °C) to eliminate any basal effect of brown fat and muscle shivering activity; mice were anesthetized using pentobarbital (120 mg/kg) at 30 °C and placed in metabolic cages with environmental enclosures at 30 °C (CLAMS-ENC). After 45–60 min, CL was subcutaneously injected (1 mg/kg) and mice were placed back in the metabolic cages for O$_2$ consumption measurements.

To assess BAT function during cold-stress conditions, the metabolic cages were pre-chilled at 4 °C, mice were again singly housed in the 4 °C cages, and O$_2$ was monitored for 7–8 h. To assess BAT histology under thermoneutral conditions, mice were housed singly in 30 °C cages for 4 weeks, afterwards mice were sacrificed and BAT was excised, weighed, and fixed for histology processing.

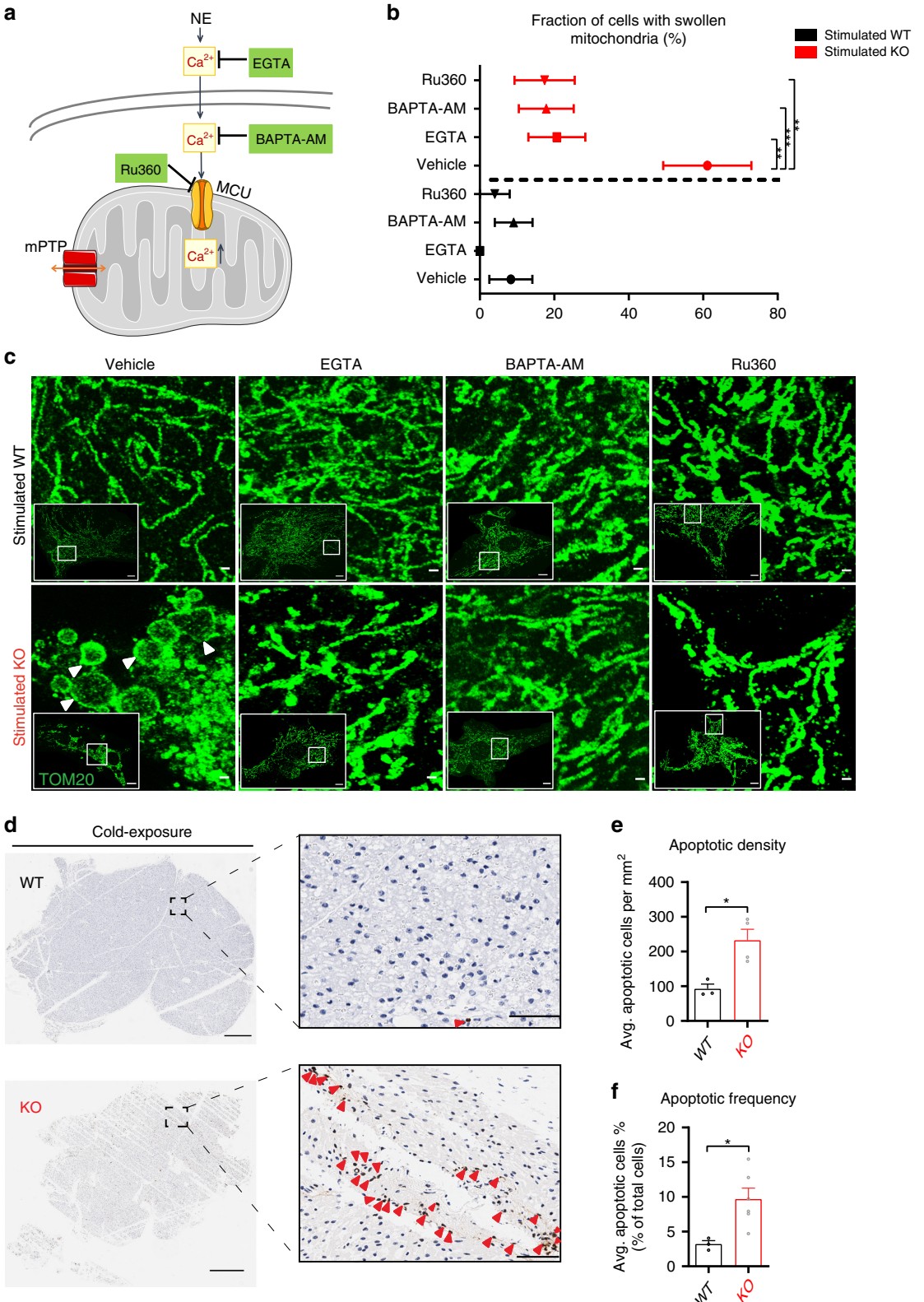

**PET/CT imaging.** Following 6–8 h of cold-stress, in vivo μPET/CT imaging was performed at the Crump Institute Preclinical Imaging Technology Center. Mice were injected with the radioactive glucose analog $^{18}$F-FDG as reported previously[46]. In brief, mice were injected via lateral tail vein with 70 μCi of $^{18}$F-FDG followed by 60 min of unconscious uptake under 1.5% isoflurane of anesthesia and cold conditions. This was followed by a static μPET/CT imaging. a Region-of-Interest analysis was conducted using AMIDE software[47] on dorsal BAT. The uptake was normalized to the liver as a reference tissue.

**Cell culture.** Primary BA were generated by differentiating preadipocytes isolated from BAT as described in detail previously[48–50]. BAT was harvested from 3 to 4-week-old WT and NCLX KO mice. The tissue was dissected from interscapular, subscapular, and cervical regions, minced, and transferred to a collagenase digestion buffer (2 mg/ml Collagenase Type II in 100 mM HEPES, 120 mM NaCl, 4.8 mM KCl, 1 mM CaCl$_2$, 4.5 mM glucose, 1.5% BSA, pH 7.4) at 37 °C water incubator under constant agitation for 30 min with vortex every 5 min. Digested tissue was homogenized and strained through 100 and 40 μm strainers. Cold

**Fig. 6 Adrenergic stimulation in NCLX-null BA leads to $Ca^{2+}$-mediated cell death in vitro and in vivo. a–c** $Ca^{2+}$ chelation and inhibition of $Ca^{2+}$ uptake prevent mitochondrial swelling in NCLX KO BA. **a** A scheme depicting points of pharmacological intervention targeting $Ca^{2+}$ entry and accumulation. EGTA (3 mM) was used to buffer extracellular $Ca^{2+}$, BAPTA-AM (25 μM) was used to chelate intracellular $Ca^{2+}$, and Ru360 (10 μM) was used to inhibit MCU and $Ca^{2+}$ entry to the mitochondria. **b** Quantification of mitochondrial swelling in NE-stimulated NCLX KO and WT BA treated with $Ca^{2+}$ chelators or Ru360 assessed by super-resolution imaging of TOM20 staining for mitochondrial outer membrane ($n = 18$–33). **c** Representative images of NE-stimulated NCLX KO and WT BA treated with $Ca^{2+}$ chelators or Ru360. Note that chelating $Ca^{2+}$ or blocking mitochondrial $Ca^{2+}$ entry was sufficient to inhibit mitochondrial swelling. Scale bar, 10 and 1 μm for the zoom images. Swollen mitochondria are indicated by the white arrows. **d–f** Cell death measured by TUNEL staining. Mice were submitted to intermittent cold-stress protocol for 5 days. BAT was harvested, fixed, and stained with TUNEL. Hematoxylin was used as a counterstaining for all nuclei ($n = 3$–6 mice per genotype). **d** TUNEL staining of BAT lobes from cold-stressed NCLX KO and WT mice. TUNEL-positive cells (brown) are indicated with red arrows. Scale bar, 500 and 50 μm for the zoom images. **e** Quantification of the average number of positive cells per cross-sectional area in each genotype. **f** Quantification of the TUNEL-positive cells as a percentage of total cells. Student's $t$-test (**e**, **f**); one-way ANOVA with Tukey's post hoc test (**b**). Data are expressed as means ± SEM. $^{ns}p > 0.05$, $^*p < 0.05$, $^{**}p < 0.01$, $^{***}p < 0.0001$. Replicates are indicated by individual dots shown for each group in all graphs. Source data are available as a Source Data file.

DMEM was added to tissue digest and centrifuged twice (the last included washing and resuspension in new DMEM at $200 \times g$ speed for 12 min at 4 °C). Finally, cell pellets (preadipocytes) were re-suspended 5 ml growth medium (DMEM supplemented with 20% newborn calf serum (NCS), 4 mM glutamine, 10 mM HEPES, 0.1 mg/ml sodium ascorbate, 50 U/ml penicillin, 50 μg/ml streptomycin) and plated in six-well plates (Corning). Cells were incubated in a 37 °C 8% $CO_2$ incubator. Twenty-four hours after isolation, the cells were washed to remove debris and the medium was replaced. Seventy-two hours after isolation the cells were lifted using STEMPro Accutase, counted, and re-plated in differentiation media (growth media supplemented with 1 μM rosiglitazone maleate and 4 nM human recombinant insulin). Cells were differentiated for 7 days and the medium was changed every other day. For transduction experiments, cells were transduced with virus in differentiation days 0–3 (see below).

For mPTP inhibition experiments, cells at the beginning of the differentiation process were co-treated with DMSO or CyPD inhibitor NIM811 at a concentration of 500 nM (Novartis).

**Virus preparation**. mCherry-GFP-FIS1(101–152) construct was a generous gift from Ian Ganley[51]; plasmid was packaged into adenoviral particles (Welgen) and was used to stain outer mitochondrial membrane. For the studies presented here, only the mCherry fluorophore was excited and recorded. 2mtGCaMP6m construct was a generous gift from Diego De Stefani[26] and the adenoviral particles were prepared as described previously[50,52].

**Measurement of oxygen consumption rate**. Ten thousand primary brown preadipocytes were plated, grown, and differentiated for 7 days on a Seahorse 24-well microplate (Agilent, Santa Clara, CA). Oxygen consumption rate of the cells was measured using a Seahorse XFe24 Analyzer, as previously described[49,53]. Normalization to cell number was done using the Operetta fluorescence microscope by counting the number of nuclei using Hoechst staining (2 μg/ml; Thermo).

Before the experiment, cells medium was replaced to assay media (DMEM modified medium without sodium bicarbonate (D5030; Sigma) with an addition of 3 mM glucose and 2 mM glutamine, followed by incubation for 60 min at 37 °C (in a non-$CO_2$ incubator) before loading into the XFe24 extracellular analyzer (Agilent). During these 60 min, the ports of the cartridge containing the oxygen probes were loaded with the compounds to be injected during the assay (50 μL/port) and the cartridge was calibrated. After steady basal consumption rates were obtained, NE was injected first to a final concentration of 1.5 μM. This was followed by injection of oligomycin (Calbiochem, San Diego, CA, US) at a final concentration of 2 μM, followed by injection of (carbonyl cyanide 4-(trifluoromethoxy) phenylhydrazone) FCCP (Sigma, C2920) at a final concentration of 2 μM, Finally, 3 μM Antimycin A (Sigma, A8674) was injected to get non-mitochondrial respiration fraction.

For mitochondrial complexes activity assessment, BAT tissue was excised, rinsed with 1% PBS, flash-frozen in liquid nitrogen, and stored at −80 °C until further analyses. BAT was homogenized in MAS buffer (70 mM sucrose, 220 mM mannitol, 5 mM $KH_2PO_4$, 5 mM $MgCl_2$, 1 mM EGTA, 2 mM HEPES) in a Teflon homogenizer. Samples were spun for 10 min at $1000 \times g$ at 4 °C to remove nuclear and membrane debris. Supernatant was used for frozen respirometry assays in a XFe96 Seahorse equipment. Homogenates (8 μg per well in 20 μl volume, minimum of 3–4 replicas per condition) were loaded in a XFe 96-well plate and centrifuge at $2000 \times g$ for 5 min at 4 °C using plate carrier rotating buckets in order to adhere mitochondrial particles to the bottom of the plate, with no brake. After centrifugation, 130 μl MAS plus cytochrome $c$ (10 μg/ml final concentration) was added per well. Respirometry assay was performed by serial injections of NADH or succinate-rotenone (1 mM or 5 mM–2 μM final concentrations, respectively) in port A; antimycin A (5 μM final concentration) in port B; TMPD/ascorbate (0.5 mM/1 mM final concentrations) in port C, and azide (50 μM final concentration) in port D. OCR were calculated as the substrate minus inhibitor-dependent rates per μg of tissue homogenate.

**siRNA preparation and transfection**. Double-stranded siRNAs used to silence NCLX expression were obtained from Ambion (Applied Biosystems) as reported before[18]. The sequence of 21 nucleotides corresponding to the sense strands used for the NCLX siRNA was AACGGCCACUCAACUGUCUtt and that for the control siRNA was AACGCGCAUCCAACUGUCUtt. siNCLX or siControl were diluted in Lipofectamine 3000 transfection reagent according to the manufacturer's instructions (Thermo). The efficiency of transfection was assessed by visualizing co-transfecting Dharmacon siGLO Green transfection particles according to the protocol provided by the manufacturer (Dharmacon, D-001630-01-05). The transfection efficiency for siNCLX delivery as determined by siGlo fluorescent marker was ~70–90%.

**High-throughput imaging**. Cells stained with Mitotracker Green (MTG) were imaged on PerkinElmer Operetta high-content wide-field fluorescence imaging system coupled to Columbus analysis software. ×40 NA objective lens was used in a single focal plane across each plate. For long-term imaging experiments, the cells were kept inside an environmental chamber at 37 °C and 5% $CO_2$ levels. The bottom of each well was detected automatically by the Operetta focusing laser, and the focal plane calculated relative to this value. MTG excitation (460–490 nm) and emission (520–550 nm) was imaged for ~100 ms with a total of 15 fields of view taken per well, with an identical pattern of fields used for every well. Modified Columbus (PerkinElmer) image analysis software was used to calculate the intensity of the fluorescent probe.

**Fluorescent $Ca^{2+}$ imaging**. Kinetic live-cell fluorescent imaging was performed to monitor $Ca^{2+}$ transients using two imaging systems. The first system consisted of an Axiovert 100 inverted microscope (Zeiss, Oberaue, Germany), Polychrome V monochromator (Till Photonics, Planegg, Germany), and a Sensi-Cam cooled charge-coupled device (PCO, Kelheim, Germany). Fluorescence images were acquired with Imaging WorkBench 6.0 software (Axon Instruments, Foster City, CA, USA). The second system consisted of an IX73 inverted microscope (Olympus) equipped with pE-4000 LED light source and Retiga 600 CCD camera. All images were acquired through a ×20/0.5 Zeiss Epiplan Neofluar objective using Olympus cellSens Dimension software.

$Ca^{2+}$ imaging was performed in BA that were grown and attached onto coverslips, mounted in a chamber that allowed perfusion of cells, and superfused with a Krebs–Ringer's solution containing (in mM): 123 NaCl, 5.4 KCl, 0.8 $MgCl_2$, 20 HEPES, 1.8 $CaCl_2$, 15 D-glucose, 2 glutamine, and 1% free-fatty acid bovine serum albumin (BSA); pH was adjusted to 7.4 with NaOH. In some experiments, $NMDG^+$ was used to replace NaCl in $Na^+$-free Krebs–Ringer's solutions and pH was adjusted to 7.4 with KCl[18]. For mitochondrial $Ca^{2+}$ measurements, cells were washed and then loaded with Rhod-2AM (1 μM) for 30 min at 37 °C. After loading cells were washed again three times followed by additional incubation of 30 min to allow for the de-esterification of the dye. Rhod-2AM was excited at 552 nm wavelength light and imaged with a 570 nm long-pass filter. For GCaMP6-mt experiments, fluorescence was acquired at 490 nm ($Ca^{2+}$-sensitive wavelength) and fluorescence was collected through a 535 nm band-pass filter.

Mitochondrial $Ca^{2+}$ response was triggered by switching the perfusion solution to Krebs-Ringer's solution supplemented by NE (1.5 μM) or ATP (100 μM). In some experiments, cells were pretreated with Forskolin (Sigma) or H-89 (Santa Cruz Biotechnology) at 50 μM and 5 μM concentrations, respectively.

For cytosolic $Ca^{2+}$ measurements, cells were loaded with 5 μM Fura-2AM (0102, TEF Labs) for 30 min at 37 °C. After loading cells were washed again three times followed by additional incubation of 30 min to allow for the de-esterification of the dye and excited with 340/380 nm wavelength light and imaged using a 510 nm long-pass filter, as described previously[54].

Traces of $Ca^{2+}$ responses were analyzed and plotted using KaleidaGraph. The rate of ion transport was calculated from each graph (summarizing an individual

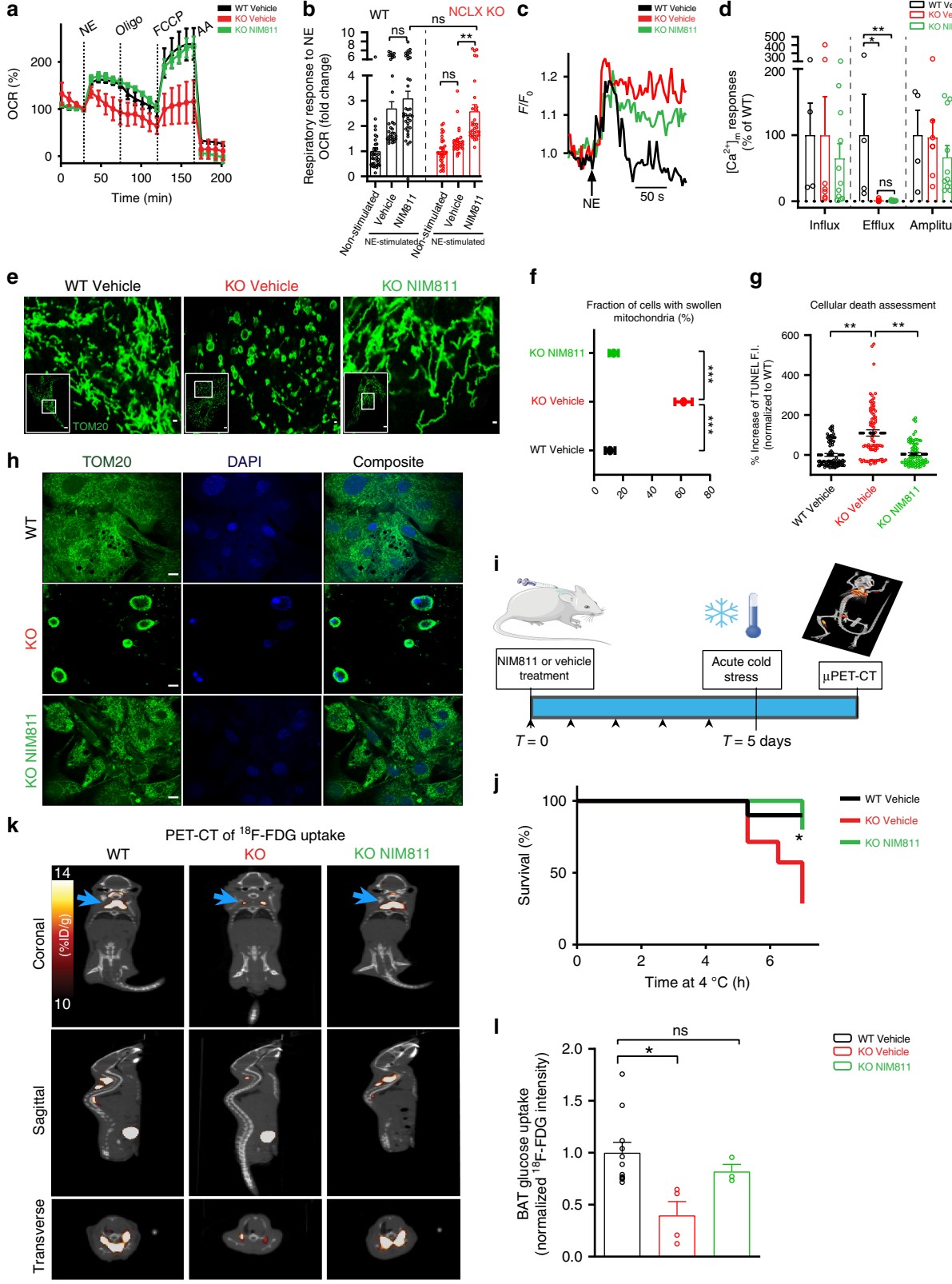

experiment) by a linear fit of the change in the fluorescence ($\Delta F$) for $Ca^{2+}$ influx and efflux over time ($\Delta F/\mathrm{d}t$). Rates from $n$ experiments (as mentioned in legends to the figures) were averaged and displayed in bar graph.

**Super-resolution microscopy**. A Zeiss LSM 880 confocal microscope with Airyscan mode was used for super-resolution imaging, with 405, 488, and 561

nm lasers and ×63 objective. DAPI was excited with 405 nm 30 mW laser, 488 nm laser was used to excite Alexa Fluor 488, Mitotracker Green and GCaMP6-mt, and 561 nm laser was used for excitation of Alexa Fluor 568, TMRE, and mCherry fluorophore (in experiments of BA infected with mCherry-GFP-Fis-1 AAV). At least 25 cells per condition were collected, the cells were individualized as areas of interest using FIJI ImageJ software (https://fiji.sc/), and further image analysis was applied on them as described below.

**Fig. 7 Inhibition of mitochondrial permeability transition in NCLX-null mice restores thermogenic function. a, b** OCR of NCLX-null and WT primary BA that were pretreated either with the mPTP inhibitor, NIM811, or with vehicle control, DMSO. **a** Representative OCR traces in BA from NCLX KO and WT pretreated with NIM811 (500 nM) or vehicle control. **b** Quantification of NE response in NCLX KO and WT BA pretreated with NIM811 or vehicle control ($n = 28$–36 per condition from $N = 8$–9 independent experiments; See Supplementary Fig. 10 for WT treatment traces). **c** Representative mitochondrial $Ca^{2+}$ transients of NCLX KO and WT BA pretreated with NIM811 or vehicle control and stimulated by NE. **d** Quantification of $Ca^{2+}$ amplitude, influx, and efflux rates ($n = 4$–11 independent experiments per condition). **e** Representative images of cells stained for TOM20 to label mitochondrial localization and assess mitochondrial swelling. Scale bar, 10 and 1 μm for the zoom images. **f** Quantifications of mitochondrial swelling in NE activated WT and NCLX KO BA pretreated with NIM811 or vehicle control ($n = 68$–118 cells). **g** Cell death in WT and NCLX KO BA pretreated with NIM811 or vehicle control after 72 h of NE stimulation. Cell death was measured in vitro by TUNEL assay ($n = 77$–83 per condition from $N = 5$–6 independent experiments). **h** Representative images of WT and NCLX KO BA after 72 h of NE stimulation. Cells were pretreated with NIM811 or vehicle control. TOM20 was used to label mitochondrial localization while DAPI was used to label the nuclei. Scale bar, 10 μm. **i** Schematic graph for in vivo NIM811 experimental procedure. Mice were pretreated for 5 days with a 50 mg/kg dose NIM811 or a vehicle introduced subcutaneously. On the day of the experiment, mice were acutely cold stressed (6–8 h) and then immediately PET-imaged with the glucose analog $^{18}$F-fluorodeoxyglucose ($^{18}$F-FDG) to measure BAT activity under 4 °C. **j** Survival curves of 8–10-week-old male NCLX KO and WT mice pretreated with or without NIM811 and cold stressed at 4 °C. Mice reaching 28 °C or lower were returned to room temperature for recovery ($n = 5$–10 mice per group). **k** Representative PET-CT images of $^{18}$F-FDG uptake in cold-stressed mice. Each image is a composition of a CT image superimposed on a PET image. In the CT image, radio-opacity is reflected in grayscale. In the PET image, $^{18}$F-FDG intensity is reflected in a hot-metal scale. BAT is marked with blue arrows. $^{18}$F-FDG uptake is presented as percent injected dose per gram (%ID/g). **l** Quantification of $^{18}$F-FDG uptake in BAT and normalized to liver uptake as a reference tissue ($n = 3$–11 mice per group). One-way ANOVA with Tukey's post hoc test (**b**, **d**, **f**, **g**, **l**); log-rank (**j**). Data are expressed as means ± SEM. $^{ns}p > 0.05$, $^{*}p < 0.05$, $^{**}p < 0.01$, $^{***}p < 0.0001$. Replicates are indicated by individual dots shown for each group in all graphs. Source data are available as a Source Data file.

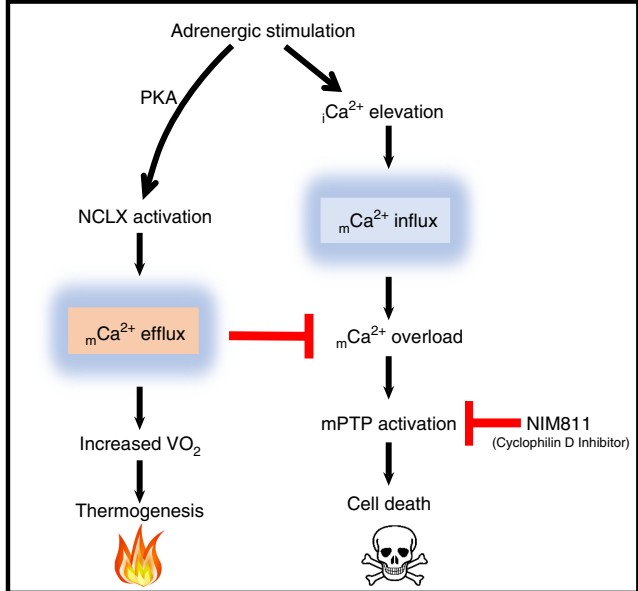

**Fig. 8 Mechanism by which brown adipocytes avoid cell death during thermogenesis.** Adrenergic stimulation of brown adipocytes leads to intracellular increase in $Ca^{2+}$ which can induce mitochondrial $Ca^{2+}$ overload and permeability transition. While in other cell types, such scenario will progress to apoptosis, in BA, adrenergic stimulation leads to activation of $Ca^{2+}$ extrusion through NCLX, thereby preventing permeability transition and cell death. Inhibition of permeability transition using the Cyclophilin D inhibitor, NIM811, can prevent cell death and restore function even in the absence of NCLX, and while $Ca^{2+}$ overload persists. $_iCa^{2+}$, intracellular $Ca^{2+}$; $_mCa^{2+}$, mitochondrial $Ca^{2+}$.

**Electron microscopy.** Primary BA were cultured on Thermanox coverslips (Electron Microscopy Sciences, 72274), on day of the experiment cells were fixed in 2.5% glutaraldehyde for 1 h. cells were then stored in 0.1 M phosphate buffer at 4 °C and processed for electron microscopic examination. Cells were postfixed in 1% osmium tetroxide for 1 h at RT, dehydrated in ethanol, and flat embedded in fresh epoxy resins. After samples processing, resin blocks were trimmed with razor blades. Semithin sections at 0.5 μm were cut using a Histo knife (Diatome). Ultrathin sections at around 70 nm were sectioned using a diamond knife (Diatome), counterstained with uranyl acetate and lead citrate, and then observed at a

JEOL 100CX transmission electron microscope (JEOL Ltd., Tokyo, Japan). For morphometric and swelling measurements, at least 185 mitochondria were analyzed per condition using FIJI ImageJ software.

**Fluorescent dyes.** Mitotracker Green (MTG) (Thermo) was used at 200 nM. Staining proceeded for 45 min followed by three washes with PBS at 37 °C. before imaging. Rhod-2AM (Thermo) was used at 1 μM; cells were stained for 30 min at 37 °C. Dye was washed three times with Ringer's buffer and re-incubated for an extra 30 min to allow de-esterification of the intracellular AM esters.

Hoechst was used at 2 μg/ml concentration (Sigma, 33258) to stain the nuclei for 30 min prior to counting nuclei for Operetta experiments for counting nuclei of Seahorse assays. DAPI was loaded at 1 mg/ml and used to stain nuclei of fixed cells on the day of the experiment. For mitochondrial membrane potential imaging, TMRE was loaded at 15 nM for 90 min with MTG co-staining, followed by wash-out before imaging. TMRE was present during imaging.

**Blue native gel electrophoresis.** Fifty micrograms protein of BAT homogenates were used for Blue Native Gel Electrophoresis for mitochondrial super complex analysis, as described previously[55]. One milligram digitonin/mg of homogenate protein was added and samples were incubated on ice for 5 min. One percent digitonin (Sigma) was dissolved in PBS by boiling and stored at 4 °C until use. Solubilized samples were centrifuged at maximal speed in a microcentrifuge for 30 min at 4 °C. Pellet was discarded and supernatant was combined with 1 μL of 2.5% Coomassie G-250. Samples were loaded into NativePAGE 3–12% Bis-Tris gel and electrophoresed at 4 °C in xCell SureLock (Novex) in constant voltage at 20 V for 60 min and 200 V for 120 min or until dye front exited the gel.

Proteins were then transferred to methanol-activated PVDF membrane in xCell SureLock in 30 V constant voltage for 1 h at 4 °C. Coomassie was completely washed off blue native blots using 100% methanol.

**Western blotting and antibodies.** After treatment, cells were rinsed three times with ice-cold PBS and scraped in an ice-cold RIPA lysis buffer (containing 50 mmol/l Tris-HCl, pH 7.5, 0.1% (w/v) Triton X-100, 1 mmol/l EDTA, 1 mmol/l EGTA, 50 mmol/l NaF, 10 mmol/l sodium β-glycerophosphate, 5 mmol/l sodium pyrophosphate, 1 mM sodium vanadate, and 0.1% (v/v) 2-mercaptoethanol), and protease inhibitors (a 1:1000 dilution of protease inhibitor mixture; Sigma P8340). The lysates were shaken for 20 min at 4 °C, centrifuged (13,500 × $g$, 15 min at 4 °C), and the supernatant was collected. Protein concentration was determined using the BCA protein assay (Pierce). Equal amounts of protein (12 μg) were mixed with 4× LDS sample buffer before running on 15% SDS-polyacrylamide gel electrophoresis and transferred onto a polyvinylidene difluoride membrane (GVS Life Sciences, 1214429) using a wet transfer system (Bio-Rad Hercules, California, US). The membranes were blocked with 5% nonfat dry milk for 1 h and then incubated with the following antibodies: Total OXPHOS Rodent WB Antibody Cocktail (1:1000, Abcam, ab110413), TOM20 (1:2000, Santa Cruz, SC-11415), UCP1 (1:1000, Abcam, ab10983), NCLX (1:1000, Santa Cruz SC-161921 and SC-161922), MCU (1:1000, Santa Cruz, SC-515930), PMCA (1:500, Santa Cruz, SC-271917), SERCA2 (1:1000, Santa Cruz SC-376235), RYR-1/2 (1:500, 34C Developmental Studies Hybridoma Bank University of Iowa).

For super complex detection, anti-Complex III Core2 UQCRC2 (1:1000, Proteintech, 14742-1-AP) and anti-Complex I NDUFA9 (1:1000, Abcam, ab188373) antibodies were used.

Antibodies were used according to the manufacturer's instructions. After overnight incubation, membranes were washed with phosphate-buffered saline containing 0.1% Tween-20 and then incubated with anti-rabbit IgG secondary antibody (Cell Signaling Technology) solution (1:5000) for 1 h or anti-mouse (1:2000) for 1 h. The membrane was again washed as above and then exposed to a chemiluminescent protein detection system (ChemiDoc, MP Imaging system, Bio-Rad).

As an alternative for beta actin, stain-free blotting was used as a normalization method according to the manufacturer's instructions (Bio-Rad). Briefly, 0.6% Trichloroethanol (Sigma, T54801) was added to the separating gel. At the end of the electrophoresis of the gel, the gel was exposed to UV for 1 min in order to conjugate tryptophans of the samples with the TCE. After the transfer, the membrane was taken to the ChemiDoc MP system, and an image of the marked proteins was taken. The normalization factor was calculated using the Image lab software, version 5.2.1 (Bio-Rad, Hercules, California, US).

**Immunostaining and immunocytofluorescence**. Cells were cultured and differentiated on quadrant dishes and fixed at 4% vol/vol paraformaldehyde (PFA) for 15 min at room temperature. After washing three times in PBS, cells were incubated in permeabilization buffer (2 μl/ml Triton X-100 and 0.5 mg/ml sodium deoxycholate in PBS, pH 7.4) for 15 min at room temperature. Subsequently, cells were blocked with 3% BSA for 1 h at room temperature. Next, cells were incubated with 1:200 primary antibody of cytochrome *c* (Abcam, ab110325) at 4 °C overnight. The next day, cells were washed in PBS and incubated with 1:200 primary antibody of TOM20 (Santa Cruz, Tom20 FL-145) at 4 °C for overnight. On the last day, cells were incubated with 1:500 Anti-Rabbit Alexa Fluor 488 (Thermo, A11008) or Anti-Mouse Alexa Fluor 568 antibodies (Thermo, A11004) for 1 h at room temperature, and samples were kept in PBS.

The in vitro TUNEL assays were carried out using the click-iT TUNEL Alexa Fluor Imaging Assay kit (Thermo, C10245), in accordance with the manufacturer's instructions.

**Image analysis**. Cytochrome *c* release was determined by quantifying integrated extra-mitochondrial FI (AU) using FIJI software and normalized per total cell area ($μm^2$). Swelling was assessed by visual inspection scoring cells for containing fragmented and spherical mitochondria with a diameter larger than 4 μm after 6 h of NE stimulation, or a diameter >2.5 μm after 4 h of NE stimulation. Cells with more than five mitochondria that were positive for these criteria were scored positive. Macros built for quantification of cytochrome *c* release and mitochondrial membrane potential analysis were used as previously described in detail[56].

**Preparation of images for print**. Representative microscopy images were transformed from the microscopy format generated by Zen Software (.CZI) to TIF images using FIJI software. Images were cropped for detail, separated into respective channels, and the window and level parameters were adjusted identically per channel in all images to emphasize the fluorescent structures in the images without manipulation of raw pixel values. Images were then inserted in PowerPoint software. In the TOM20 and cytochrome *c* co-staining experiments photo-correction was used to enhance the brightness of all images by increasing it by 40% and lowering the contrast by 40% for illustrative purposes.

Western blot images were also transferred to PowerPoint. When Brightness/Contrast were applied, it was done equally to control and experimental groups including all blot area.

For the mCherry-Fis1 experiments and subsequent super-resolution microscopy experiments assessing mitochondrial swelling and architecture, the aim is to show morphological changes. Therefore, brightness and contrast for each condition were optimized using FIJI taking into account the image's histogram (window and level parameters were not identical).

**Histology and immunohistochemistry analysis**. For in vivo cell death assessment in BAT, mice were continuously cold-stressed for 4 days, 6–8 h each day. On the last day, mice were sacrificed and iBAT was excised and fixed overnight for 16 h in 10% buffered formalin and then placed in 70% ethanol. Tissues were then processed and embedded by Translational Pathology Core Laboratory (TPCL) at UCLA and TUNEL ApopTag staining was performed on them according to the following the kit instructions (Millipore, S7101); hematoxylin was used for counterstaining. For additional controls of death assessment, iBAT from mice housed at room temperature of 25 °C and thermoneutrality (30 °C) were excised, fixed, and processed by TPCL. The slides were stained with Cleaved Caspase-3 Antibody (Cell Signaling, 9664) and hematoxylin was used as a counterstaining. Slides were then scanned onto an Aperio ScanScope AT at ×20 magnification (Aperio Technologies, Inc., Vista, CA). Digital slides were blindly analyzed with QuPath software in order to determine percent and area of TUNEL-positive cells. For the calculation of area with dead cells (apoptotic nuclei density), analysis was done blindly and data exclusion criteria were applied to all samples where microscopy fields contained more than 15% of area without cells in the BAT lobe to avoid artefacts introduced by calculating areas lacking cells[57].

**Statistics**. All data are expressed as mean ± SEM. Statistical analysis was performed using Prism 6.0 (GraphPad Software) and two-tailed Student's *t*-test was used for two groups. Two-way ANOVA followed by Sidak's test was used for multiple comparisons involving two groups and one-way ANOVA with post hoc Tukey's test was used for multiple comparisons.

For in vivo studies including μPET/CT imaging, histological sectioning, staining, and analysis, investigators were blinded from mouse genotype and an ear-tagging system enabled unbiased data collection. In vivo experiments involved core facilities that were not familiar with the study, limiting biased results.

**Reporting summary**. Further information on research design is available in the Nature Research Reporting Summary linked to this article.

## Data availability

The dataset generated during the study, including imaging data, are available on reasonable request. The source data underlying Figs. 1–7 and Supplementary figures are provided as a Source Data file. The exact *n* values for each experimental group in Figs. 1–7 are included in the source data file.

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

## Acknowledgements

The authors would like to thank those who contributed helpful discussions, insight, and support of the research, including Drs. Kiana Mahdaviani, Ilan Benador, Eleni Ritou, Evan Taddeo, Linsey Stiles, Nour Alsabeeh, Anton Petcherski, Fernanda Cerqueira, Jennifer Ngo, and Kyle Trudeau from the Shirihai lab and Drs. Marko Kostic, Soumitra Roy, Tatiana Tkatch, and Tomer Katoshevski from Sekler lab. The authors would like to thank Drs. Assaf Rudich, Alicia Kowaltowski, Dani Dagan, Jan Nedergaard, Barbara Cannon, Barbara Corkey, Marc Prentki, Michal Hershfinkel, Ajit Divakaruni, Ehud Ohana, and Daniel Khananshvili for their helpful advice and discussions. We thank Dr. Jason T. Lee and Charles Zamilpa at UCLA's Crump Imaging Technology Center for assistance with PET/CT imaging of the mice. We thank Translational Pathology Core Laboratory at UCLA's DGSOM for assistance with histology samples preparation and processing and we thank Dr. Chunni Zhu from the electron microscopy core at the Brain Research Institute, UCLA for her assistance with electron microscopy acquisition and sample preparation. Graphical illustrations were created using Servier Medical Art templates, which are licensed under a Creative Commons Attribution 3.0 Unported License. E.A.A. is an Azrieli fellow and was supported by the Kreitman predoctoral scholarship from Ben-Gurion University and the Israeli Council for Higher Education fellowship. O.S.S. is funded by National Institutes of Health grants RO1 DK35914, R01 DK56690, and R01 DK074778. I.S. is funded by Israel Science Foundation (ISF, 1424/17), ISF-China (1210/14), and German-Israeli Project Cooperation (DIP, SE2372/1-1). M.L. is funded by the DRC UCLA/UCSD Pilot grant (NIH P30 DK063491) and the Department of Medicine Chair at UCLA.

## Author contributions

E.A.A. designed and performed experiments, analyzed the data, and wrote the manuscript. O.S.S. and I.S. helped design the study and supervised manuscript writing. A.E.J. designed and performed mouse experiments and analyzed PET/CT data. M.V. helped managing the mice colony and breedings. R.A.-P. performed biochemical assays for complexes activity. N.A.M. built image analysis macros and helped with microscopy. M.T. performed BAT isolation and culture. M.S. helped operating the metabolic cages. G.L. and M.F.O. helped with data interpretation, and manuscript editing. M.L. helped with experiments design, data interpretation, and manuscript editing. All authors read and approved the final version of the manuscript.

## Competing interests

The authors declare no competing interests.
