## [Peer Review File · Nature Communications]

Reviewers' comments:

Reviewer #1 (Remarks to the Author):

The paper by Essam A. Assali and co-workers show the pivotal role of NCLX in regulating the thermogenic metabolic function of the Brown Adipose Tissue (BAT) by adrenergic stimulation, preventing the activation of mPTP and cells death. Overall the manuscript is well written and gives a good summary of the role of NCLX that act as an anti-apoptotic mechanism by inhibiting mitochondrial Ca²⁺ overload. However, following comments and issues should be considered for further improving and optimizing this manuscript:

Major point:

Figure 1B: The authors affirm that mitochondrial Ca²⁺ uptake is not dependent on Na⁺ in BA. The reported traces of mitochondrial Ca²⁺ responses are not in line with this consideration. A marked difference in the amplitude of mitochondrial Ca²⁺ uptake is evident between the two reported traces.

Figure 1 and 2: The representative traces of mitochondrial Ca²⁺ responses reported in figure 1 and 2 of same experimental conditions differ in term of amplitude and mitochondrial Ca²⁺-uptake rate. Uniform all the representative traces. Indeed, I suggest quantifying the NE-induced mitochondrial Ca²⁺ response also with the ratiometric fluorescent probe mt-GCaMP, to confirm that there are no changes in amplitude and mitochondrial Ca²⁺ uptake.

Figure 1E: The authors show that ATP blocks the mitochondrial Ca²⁺ efflux in BA. How to change the cell viability when the cells are exposed to ATP? Especially considering that the block of mitochondrial Ca²⁺ efflux induces Ca²⁺ overload and mPTP opening.

Figure 1I: We suggest to uniform all the histograms reported.

Figure 2J: The graph shows no difference in OCR in basal conditions between WT and NCLX-KO. Data would be more complete if authors measured the mitochondrial membrane potential in WT and NCLX-KO BA in basal conditions and in response to NE.

Figure 3B: If NCLX is pivotal to thermogenesis in BA, why the basal temperature is not dramatically changed in NCLX-null mice? And if so, how changes the basal temperature between WT and NCLX-null mice? Which is the compensatory mechanism recruited?

Figure 4: The authors affirm that mitochondrial physiology results unchanged between WT and NCLX-null mice. The data reported in figure 4 are weak; we suggest to integrate with electronic microscopy acquisitions and morphometric analysis. I also recommend checking the expression levels of different mitochondrial proteins, including MCU complex components such as MICU1, as indicated in the discussion section.

Figure 5C: Controversial is the data reported in figure 5C, where the authors showed an increase in mitochondrial fragmentation in NCLX-null mice. This results in contrast with data represented in figure 4 and with the mitochondrial Ca²⁺ response of figure 2.

Figure 6: In figure 6 authors show that NIM811 rescues the respiration in NCLX-null cells in vitro and the survival and thermogenic function in vivo. Although this, it is unclear how the mitochondria

dissolve the NE-induced mitochondrial Ca²⁺ overload in NLCX-null BA. How changes mitochondrial Ca²⁺ signaling and structure in NIM811-treated NLCX-null BA in response to NE? Who dissipates the NE-induced mitochondrial Ca²⁺ overload in NLCX-null BA if mPTP is blocked?

Authors assert that the mitochondrial Ca²⁺ overload is not deleterious per se for mitochondrial bioenergetics as long as mPTP remains closed. To suffrage, this statement, can the authors compare cell viability between WT and NCLX-null BA treated with NIM811 in response to NE at different time points?

Minor points:

- Replace [Ca²⁺]_{mito} with [Ca²⁺]_m
- The data in supplementary figure 1 a-d are needless considering the off-target effects of lack of Na⁺ on metabolic function, as suggested by authors. An imbalance in basal respiration in wt cells cultured in NMDG⁺-medium has been detected in the basal condition.
- Change reference n. 18 with newer manuscripts that describe the uniporter structure such as 30143745.

In the chapter Materials and Methods in the paragraph:

- Cell culture: There is a repetition of the same concept
“collagenase digestion buffer at 37°C under constant agitation for 30 min. Collagenase digestion was performed in 37°C water incubator under constant agitation for 25 min with vortex agitation every 5 min.”
- Measurement of Oxygen Consumption rate correct (50 μ L/port)
- Western Blotting and antibodies: Commas not needed in the dilutions value “1:1,000 dilution... ..solution (1:5,000) for 1 h or anti-mouse (1:2,000)”
- Immunostaining and Immunocytofluorescence correct (2 μ L/mL Triton X-100)

Reviewer #2 (Remarks to the Author):

This manuscript by Assali et al described that NCLX, a mitochondrial Na⁺/ Ca²⁺ exchanger, was required for the mitochondrial calcium efflux and functioned to prevent opening of mitochondrial permeability transition pore in response to adrenergic activation in brown fat. The authors used both in vitro and in vivo systems to demonstrate that NCLX null brown adipocytes displayed mitochondrial calcium overload in response to adrenergic stimulation, leading to mitochondrial

swelling and cell death. Overall, the findings of NCLX as a key regulator of mitochondrial calcium homeostasis upon adrenergic stimulation are intriguing. However, there are flaws in experimental designs and interpretations. Here are some areas that this manuscript could be strengthened.

1. The use of the whole-body NCLX KO mice in assessing the role of this mitochondrial protein in brown fat-mediated thermogenesis and survival was problematic. Because maintaining mitochondrial calcium homeostasis is key to maintaining proper cellular function, ablation of NCLX would be expected to impact many other tissues/organs besides brown fat. Thus, the observed defects of thermogenesis, VO₂ consumption and survival rate in response to cold challenge need to be interpreted in the context of collective effects from multiple tissues/organs. In addition, what are the levels of NCLX expression across multiple tissues? Such information would be important to determine the contribution of different tissues to the overall phenotype of the knockout animals.

2. The authors showed NCLX KO mice displayed defects after NE stimulation and/or cold challenge, but it was unclear whether the KO animals exhibited any phenotype at basal conditions (i.e., untreated, room temperature or thermoneutral temperature). It is important to clarify whether the observed phenotypes were solely triggered or accelerated by adrenergic stimulation.

3. Since intracellular calcium molecules are dynamically cycling among cytosols, ER, and mitochondria, it is important to assess whether calcium dynamics are altered in the cytosol or ER in the siNCLX and NCLX KO cells. In addition, cytosolic calcium handling machineries, such as SERCA or RyR2, have been implicated in brown/beige fat thermogenesis (e.g., Ikeda et al., Nat Med 2018), thus, the authors also need to evaluate if these calcium handling systems are altered in the NCLX KO or KD cells

4. In Figure 4, the authors spent almost whole figure (Fig. 4a-4g) to address the decrease of maximal VO₂ after acute cold or CL treatment was not due to change of UCP1 or other mitochondrial proteins. However, one wouldn't expect 6 hrs of cold or CL treatment could lead to significant changes at the protein levels. What's important and yet missing in this figure was the assessment of changes in mitochondrial activity and/or changes in post-translational modifications of mitochondrial proteins. In Fig. 4h, in addition to the results of the quantification of spare respiration capacity, please also show the original seahorse curve.

5. The data presented in Fig. 5d regarding mitochondrial swelling was not very convincing. EM images of mitochondrial ultrastructure would be an alternative approach to demonstrate morphological changes of mitochondria in the NCLX KO cells.

6. In Fig. 5a and 5b, the authors showed more cytochrome c release from mitochondria in KO cells using immunofluorescent staining method. These data need to be confirmed by Western blots using samples from subcellular fractionation.

7. Was there any change in tissue size or weight of BAT observed in the NCLX null mice given the fact that the KO mice displayed increased cell death in BAT (Fig. 5e-5h)? These data would be important for the interpretation of the data shown in Fig. 6f. If the KO mice had a smaller BAT mass, it would be conceivable to observe less FDG uptake in BAT region. If that was the case, it might be prudent to normalize the FDG uptake data to BAT mass.

8. In the discussion, the authors concluded that “Mechanistically, this is the first study that shows a physiological role of PKA-regulated mitochondrial Ca²⁺ extrusion via NCLX activation”. This was apparently overstated because as the authors noted in the Introduction that it has already been shown that PKA-mediated phosphorylation of NCLX rescued mitochondrial Ca²⁺efflux in depolarized neurons lacking PINK1.

Reviewer #3 (Remarks to the Author):

In this manuscript, Assali et al. investigate the role of mitochondrial sodium/calcium exchanger (NCLX) in brown fat function, using cultured cells and mice. The authors show that loss of NCLX (1) leads to calcium accumulation in the mitochondria after adrenergic stimulation, (2) impairs cold-induced thermogenesis (3) causes cell death after adrenergic stimulation. The contribution of mitochondrial calcium signaling to brown fat physiology has not been well understood, and represents a topic of great interest to the fields of mitochondrial biology, calcium signaling and metabolisms. Even though the authors convincingly show that NCLX is important for non-shivering thermogenesis in mice, given that this is a whole body knockout, it's not possible to attribute the observed phenotype to brown fat-specific NCLX function. The authors show the importance of NCLX in protecting brown fat cells in response to adrenergic stimulation, but the finding that NCLX protects cells from PTP-mediated cell death is not novel and has been shown in the heart by Luongo et al (PMID: 28445457). In addition, PKA-regulation of NCLX function in brown adipocytes needs further experimental support.

Major points:

1. Loss of NCLX in adult mice in the heart was shown to be lethal (PMID: 28445457). It is very surprising the NCLX knockout animals survive. Western blot in Figure 3d shows that there is still detectable NCLX protein in the KO animals. This should be discussed in the manuscript.
2. One of the main claims in this manuscript is PKA regulation of mitochondrial calcium efflux, via NCLX. However, no data that place NCLX and PKA on the same pathway are presented here. What happens to mitochondrial calcium efflux if PKA is activated in NCLX KO cells (by forskolin addition for example)? The authors refer to a previous paper by Dr. Sekler that shows phosphorylation of NCLX by PKA. The previous study was conducted in neurons, is the same signaling pathway conserved in brown adipocytes? Does PKA have NCLX-independent effects on mitochondrial calcium cycle in

adipocytes? These questions should be addressed to strengthen the author's argument that the PKA regulates NCLX function in BAT.

3. Ikeda et al. showed that calcium cycling is important in beige fat for thermogenesis, independent of UPC-1 (PMID: 29131156). It is reasonable to think that NCLX may be important for calcium cycling in this tissue. The thermogenesis phenotype observed in NCLX KO mice used in this study could be a combination of brown and beige fat phenotypes, and not only due to brown fat. In addition, Jackson Labs reports that the NCLX KO mouse strain used in this study has "decreased circulating glucose level" (mouse strain 026242). These are all confounding factors while determining the BAT-specific effects of NCLX loss. Using a conditional NCLX mouse model may be beyond the scope of this study, but all these caveats should be addressed in the discussion.

4. I strongly disagree with the author's conclusion that BAT dysfunction should be the main reason for the observed phenotype. β -3 stimulation experiments show a defect in BAT, but do not exclude defects in other tissues.

Minor points:

1. The manuscript will benefit from editing for spelling and grammar. Examples:

However, for TCA cycle to meet up with the energy demand... should be:

However, for TCA cycle to meet/ keep up with the energy demand

A recent study showed that that PKA-mediated...: delete "that"

There are many others.

2. The figures should also be edited, especially the fonts. Some fonts are stretched out, font sizes and scale bars are variable in figures. In addition, different scale bars are used for similar experiments (monitoring mitochondrial calcium for example, in figures 1a and 1b), making it difficult to compare the data.

3. NMDG+ treatment of cells is likely to affect a lot of different cellular processes. The authors should use NCLX inhibitor CPG37157 in experiments shown in Fig1 to assess the effects of NCLX inhibition more specifically.

4. In the discussion the authors state "Remarkably, this study shows that mitochondrial Ca²⁺ overload is not deleterious per se for mitochondrial bioenergetics as long as the mPTP remains closed". I agree that this would be an interesting finding, however, the authors do not show that there's still mitochondrial calcium overload after NIM811 treatment of mice. The authors should look at mitochondrial swelling and calcium overload after NIM811 treatment.

We thank all the reviewers for their constructive criticism and taking the time to give us valuable feedback on our manuscript. Over the last 11 months, we performed multiple new *in vitro* and *in vivo* experiments to fully address the concerns and suggestions that have been raised as detailed below. We have provided responses (black text) to each of the reviewers' questions and comments (*blue italic text*). We hope the reviewers will agree with us that our revised manuscript is vastly improved and is now suitable for publication in *Nature Communications* journal. Thanks again for your devoted time and expertise.

Reviewers' comments:

Reviewer #1 (Remarks to the Author):

The paper by Essam A. Assali and co-workers show the pivotal role of NCLX in regulating the thermogenic metabolic function of the Brown Adipose Tissue (BAT) by adrenergic stimulation, preventing the activation of mPTP and cells death. Overall the manuscript is well written and gives a good summary of the role of NCLX that act as an anti-apoptotic mechanism by inhibiting mitochondrial Ca²⁺ overload. However, following comments and issues should be considered for further improving and optimizing this manuscript:

Response: We thank the reviewer for his/her thoughtful compliments and the constructive critique of our work. We hope our new set of experiments and data below are convincing enough to address your points.

Major point:

Figure 1B: The authors affirm that mitochondrial Ca^{2+} uptake is not dependent on Na^{+} in BA. The reported traces of mitochondrial Ca^{2+} responses are not in line with this consideration. A marked difference in the amplitude of mitochondrial Ca^{2+} uptake is evident between the two reported traces.

Response: We thank the reviewer for drawing our attention to this important point. We provide new calcium traces in the **New Figure 1B**. we uniformed and updated the traces. Our aim in this panel is to show that NE drives a Na^{+} -dependent Ca^{2+} efflux without interfering with Ca^{2+} influx. Moreover, using a genetically-encoded mitochondrial calcium sensor in WT and NCLX KO BA, confirms that mitochondrial calcium uptake does not change under these conditions. Thus, the differences observed in amplitude were just reflecting the previously described heterogeneity in traces when using the Rhod-2 dye (please see point 2 below).

Figure 1 and 2: The representative traces of mitochondrial Ca^{2+} responses reported in figure 1 and 2 of same experimental conditions differ in term of amplitude and mitochondrial Ca^{2+} -uptake rate. Uniform all the representative traces. Indeed, I suggest quantifying the NE-induced mitochondrial Ca^{2+} response also with the ratiometric fluorescent probe mt-GCaMP, to confirm that there are no changes in amplitude and mitochondrial Ca^{2+} uptake.

Response: Thank you for raising this important point, we address this issue in the **NEW Supplementary Figure 1** (and pp. 6 of the results section in the clean revised version). Indeed, the heterogeneity in the response and amplitude of mitochondrial Ca^{2+} uptake measured using Rhod-2 is known and well-characterized in different systems and cell types (e.g. Kostic et al. 2015; Kostic et al. 2018; Nita et al. 2014 and others). To address the reviewers' concern, we now measured mitochondrial calcium using GCaMP6-mt delivered via viral transduction. The measurements confirm the absence of significant changes in mitochondrial calcium influx (and amplitude) induced by NCLX deletion and that the difference in amplitude was a result of the known heterogeneity when using Rhod-2 dye.

Figure 1E: The authors show that ATP blocks the mitochondrial Ca^{2+} efflux in BA. How to change the cell viability when the cells are exposed to ATP? Especially considering that the block of mitochondrial Ca^{2+} efflux induces Ca^{2+} overload and mPTP opening.

Response: We thank the reviewer for raising this interesting point. In the revised version we have clarified the purpose of the utilization of ATP as an experimental Ca^{2+} entry inducer. ATP is frequently used to raise cytosolic and mitochondrial calcium in most cell types as it activates the purinergic receptor. Numerous studies showed that activation of the purinergic receptors does not lead to cell death, but can activate physiological responses such as cell migration, exocytosis and others (Jiang et al. 2017). The purpose of this experiment was only to demonstrate that NCLX and mitochondrial Ca^{2+} efflux are not activated in the absence of PKA signaling. In addition, the duration of the ATP-dependent mitochondrial Ca^{2+} response in our experiment was short and insufficient to trigger cell death. Moreover, we think that for mPTP opening in intact cells, we need a combination of depolarization effect, free-fatty acids (FFA) or ROS in addition to Ca^{2+} overload which ATP does not provide in this setting of non-NE activated BA.

Figure 1I: We suggest to uniform all the histograms reported.

Response: Thank you for raising this important point, we have re-analyzed and edited all the Ca^{2+} responses and histograms according to the reviewer's suggestion.

Figure 2J: The graph shows no difference in OCR in basal conditions between WT and NCLX-KO. Data would be more complete if authors measured the mitochondrial membrane potential in WT and NCLX-KO BA in basal conditions and in response to NE.

Response: Following the reviewer's advice we have monitored the basal mitochondrial membrane potential (please see the **NEW Figure 4** and pp. 9) using TMRE. To control for focal plane, we normalized the values to Mitotracker Green (MTG), a membrane potential independent dye. Our results show the absence of significant changes between WT and NCLX KO BA in mitochondrial membrane potential at resting (non-stimulated) conditions.

Mitochondrial membrane potential analysis in non-stimulated NCLX KO and WT BA:

Figure 3B: If NLCX is pivotal to thermogenesis in BA, why the basal temperature is not dramatically changed in NCLX-null mice? And if so, how changes the basal temperature between WT and NCLX-null mice? Which is the compensatory mechanism recruited?

Response: Thank you for raising this point, we clarified this issue now in the revised manuscript. In room temperature (22-25 C), there is minimal activation of BAT, which means that BAT mitochondria show low rates of calcium import and thus of mitochondrial calcium levels. However, upon acute exposure of mice to cold, the

adrenergic response is sudden and with high intensity; activating a large influx of calcium to the mitochondria due to acute NE-stimulation via the sympathetic nervous system. This is when NCLX is essential to prevent mitochondrial calcium overload, as the acute nature of cold exposure activates BAT function immediately. In addition, this is not the only mouse model behaving in this way. The mice KO for UCP1 do not show defects at room temperature but only after acute cold exposure, further demonstrating that there is a specific molecular machinery in BAT that is essential to execute and sustain a response to acute cold exposure. This text has been added to the results and discussion sections (pp.7-8 and 13).

Figure 4: The authors affirm that mitochondrial physiology results unchanged between WT and NCLX-null mice. The data reported in figure 4 are weak; we suggest to integrate with electronic microscopy acquisitions and morphometric analysis. I also recommend checking the expression levels of different mitochondrial proteins, including MCU complex components such as MICU1, as indicated in the discussion section.

Response: Following the reviewer's advice we are providing new data in the **NEW Figure 4** (panels f,g,h,i,j and pp.9 of the results section) to expand the mitochondrial characterization in the NCLX-null BA. In this figure, we provide analyses of mitochondrial membrane potential, circularity, aspect-ratio and roundness parameters, under basal conditions for both WT and NCLX-null BA. Furthermore, we are providing representative images of high resolution in the same figure (panel f). These analyses reveal minor trends in mitochondrial membrane potential as well as morphology.

In addition, as for MICU1, we have made numerous attempts to quantify changes in MICU1 using WB analysis. However, consistent with many other personal reports of people in the field, the quality of the antibody commercially available is poor. Although we were successful in viewing bands at the anticipated molecular weight, we do not feel comfortable with the validity of these blots due to the multiple non-specific bands that we could not remove by various troubleshooting procedures (different membrane blocking agents, other membrane types, different boiling temp. for the samples). Therefore, we prefer to avoid any conclusions regarding the changes in this protein and other proteins of the uniporter complex. Attached below a blot for MICU1 in BA.

Figure 5C: Controversial is the data reported in figure 5C, where the authors showed an increase in mitochondrial fragmentation in NCLX-null mice. This results in contrast with data represented in figure 4 and with the mitochondrial Ca²⁺ response of figure 2.

Response: We thank the referee for raising this point. We address this comment both with a NEW data as well as clarification of the different parameters assessed in figure 4 as compared to figure 5. In the **NEW Figure 4**, we provide morphometric analysis that allows for quantification for mitochondrial fragmentation. This new analysis, demonstrates that in the absence of NE-stimulation, comparison of WT and NCLX KO BA shows a minor (~1%) increase in the level of fragmentation in the NCLX KO BA, this is described in Figure 4 and pp.9 and presented in the previous response. Figure 5 focuses on mitochondrial swelling and this analysis showed a large increase of mitochondrial swelling following NE-stimulation of KO BA; while WT BA showed only minor change that did not reach statistical significance. Overall, our new set of data indicate that NCLX-null mitochondria are largely intact and functional at basal state compared to extensive mitochondrial swelling and rupture following cold or NE exposure as shown in the **New Figure 5** and described in pp.10-11.

Figure 6: In figure 6 authors show that NIM811 rescues the respiration in NCLX-null cells in vitro and the survival and thermogenic function in vivo. Although this, it is unclear how the mitochondria dissolve the NE-induced mitochondrial Ca²⁺ overload in NCLX-null BA. How changes mitochondrial Ca²⁺ signaling and structure in NIM811-treated NCLX-null BA in response to NE? Who dissipates the NE-induced mitochondrial Ca²⁺ overload in NCLX-null BA if mPTP is blocked?

Response: Thank you for pointing out that our hypothesis was not presented clearly enough. The surprising discovery in this study is that Ca²⁺ overload does not have a detrimental consequence if the mPTP in BAT is inhibited. In the **New Revised Figure 6**, we provide new data further supporting this discovery showing a similar mitochondrial Ca²⁺ overload pattern in the NIM811 pretreated NCLX KO BA (**New Figure 6c** and pp.12). Therefore, although NIM811 does not prevent Ca²⁺ overload, it prevents the activation of permeability transition in response to calcium overload. The opening of the

mPTP as documented by many studies, and presented here in BAT, activates cell death pathway (**New Figure 5** and pp. 10-11). Here we propose that dissociating this pathway from the mitochondrial Ca^{2+} overload is sufficient to maintain tissue viability and rescue its bioenergetics function in BAT, despite the Ca^{2+} overload. We emphasize and relate to this issue in the discussion (pp. 14).

Authors assert that the mitochondrial Ca^{2+} overload is not deleterious per se for mitochondrial bioenergetics as long as mPTP remains closed. To suffrage, this statement, can the authors compare cell viability between WT and NCLX-null BA treated with NIM811 in response to NE at different time points?

Response: Following the reviewer's suggestions we have carried this set of experiments and now appear in the **NEW Figure 6** (pp.12 of the results section). We found that NIM811 indeed was effective in rescuing KO BA after NE-stimulation by performing *in vitro* TUNEL assay of WT and NCLX KO primary BA pretreated with NIM811 or a vehicle control. Moreover, we also imaged these cells after 72 h using DAPI for nuclear staining and TOM20 for mitochondrial labeling. Our results in the **New Figure 6** suggest that NIM811 is sufficient to prevent apoptosis in NCLX KO BA.

72 h after NE-stimulation:
TOM20

DAPI

Composite

WT

KO

KO NIM811

Minor points:

- Replace $[Ca^{2+}]_{mito}$ with $[Ca^{2+}]_m$. **Response:** we did that.
- The data in supplementary figure 1 a-d are needless considering the off-target effects of lack of Na^+ on metabolic function, as suggested by authors. An imbalance in basal respiration in WT cells cultured in NMDG+-medium has been detected in the basal condition.

Response: We agree with the reviewer that the NMDG+ experiments are an extreme model and replacement of Na⁺ may also have additional off-targets. However, the purpose of presenting this data in the paper is to put the manuscript in a historic perspective, starting in reproducing the experiments that first demonstrated a possible role of mitochondrial Na⁺/Ca²⁺ exchange in thermogenesis, albeit using techniques that are non-specific and non-physiological (Al-Shaikhaly et al. 1979). Therefore, we feel that those studies are a good starting point and deserve recognition due to their pioneering role in linking mitochondrial Ca²⁺ to BAT activation. Please note that the specific role of NCLX is further supported by our *molecular* results using siNCLX and NCLX KO BA as demonstrated in Figures 1, 2 and New Supplementary Figure 1.

- Change reference n. 18 with newer manuscripts that describe the uniporter structure such as 30143745.

Response: Thanks for drawing our attention to this reference, it is now added.

In the chapter Materials and Methods in the paragraph:

- Cell culture: There is a repetition of the same concept. “collagenase digestion buffer at 37°C under constant agitation for 30 min. Collagenase digestion was performed in 37°C water incubator under constant agitation for 25 min with vortex agitation every 5 min.” **Response:** Corrected.
- Measurement of Oxygen Consumption rate correct (50 μ L/port). **Response:** corrected.
- Western Blotting and antibodies: Commas not needed in the dilutions value “1:1,000 dilution... ..solution (1:5,000) for 1 h or anti-mouse (1:2,000)”. **Response:** corrected.
- Immunostaining and Immunocytofluorescence correct (2 μ L/mL Triton X-100).

Response: corrected.

We thank the reviewer again for drawing our attention to these points, we corrected all of the typos and added the suggested reference.

Reviewer #2 (Remarks to the Author):

This manuscript by Assali et al described that NCLX, a mitochondrial Na⁺/ Ca²⁺ exchanger, was required for the mitochondrial calcium efflux and functioned to prevent opening of mitochondrial permeability transition pore in response to adrenergic activation in brown fat. The authors used both in vitro and in vivo systems to demonstrate that NCLX null brown adipocytes displayed mitochondrial calcium overload in response to adrenergic stimulation, leading to mitochondrial swelling and cell death. Overall, the findings of NCLX as a key regulator of mitochondrial calcium homeostasis upon adrenergic stimulation are intriguing. However, there are flaws in experimental designs and interpretations. Here are some areas that this manuscript could be strengthened.

Response: We thank the reviewer for his/her time reading the manuscript and the careful critique. We believe your suggested experiments improved the quality of the manuscript and the impact of our story.

1. The use of the whole-body NCLX KO mice in assessing the role of this mitochondrial protein in brown fat-mediated thermogenesis and survival was problematic. Because maintaining mitochondrial calcium homeostasis is key to maintaining proper cellular function, ablation of NCLX would be expected to impact many other tissues/organs besides brown fat. Thus, the observed defects of thermogenesis, VO₂ consumption and survival rate in response to cold challenge need to be interpreted in the context of collective effects from multiple tissues/organs. In addition, what are the levels of NCLX expression across multiple tissues? Such information would be important to determine the contribution of different tissues to the overall phenotype of the knockout animals.

Response: We fully agree that we cannot overrule additional effects related to other cell systems and these issues are extensively discussed in our revised discussion (pp. 14). However, please note that based on the large and independent *in vitro* and *in vivo* experiments and the following findings and evidences presented in the previous and revised version, we assert that there is a major role for NCLX in BAT-mediated uncoupled respiration and thermogenic function:

- 1) Mitochondrial Ca²⁺ efflux and its PKA dependence as well as uncoupled respiration is impaired in cultured NCLX-null BA.
- 2) Following β₃-adrenergic agonist injection, an approach that increases mouse oxygen consumption by selectively activating BAT thermogenesis, we see a major lack of a proper increase in oxygen consumption in NCLX-null mice.
- 3) PET scans for labeled glucose (18F-FDG) and measurements of its uptake show a robust cold-dependent metabolic activity in the WT but not the KO iBAT tissue.

- 4) NIM811 fully restores BAT metabolic activity measured *in vivo* by 18F-FDG uptake in NCLX KO mice.

Thus, although we cannot overrule a role of NCLX in thermogenesis stemming from other tissues (such as muscle); we believe that the large and independent body of experiments supports a leading role of NCLX in controlling BAT-mediated uncoupled thermogenesis. Please see our revised discussion relating to the limitations of our *in vivo* model (pp.14).

As for the expression levels of NCLX in different tissues, we previously analyzed NCLX tissue distribution (see Palty et al. 2004). Interestingly, we got high levels of NCLX in pancreas and stomach and relative medium to low expression levels were present in the heart, brain, skin, smooth muscle, and spleen tissues. However, please note that the NCLX activation mode by NE through a PKA-mediated activity is the critical step in the case of BAT according to what we show here in various different ways. NCLX expression is not expected to change during a time interval of tens of seconds to induce NE-dependent mitochondrial Ca^{2+} efflux.

2. The authors showed NCLX KO mice displayed defects after NE stimulation and/or cold challenge, but it was unclear whether the KO animals exhibited any phenotype at basal conditions (i.e., untreated, room temperature or thermoneutral temperature). It is important to clarify whether the observed phenotypes were solely triggered or accelerated by adrenergic stimulation.

Response: Thank you for drawing our attention to this very important point. In the revised version we analyzed thoroughly the phenotype of the NCLX-null mice at basal temperature. We show that at room temperature the mice core body temperature is unaffected, this is presented in the ***New Supplementary Figure 4 and main Figure 3*** (pp.7-8). Furthermore, we provide analysis in the same main figure that compares core body temperature of heterozygous and WT mice and this did not find any differences either. This has been analyzed again for females and males presented in ***Supplementary Figures 5 and 6*** and in both cases no differences were found. In addition, we now provide analysis of NCLX KO iBAT weight mass and show that it is similar to that of WT mice. as further support that there is no atrophy in the NCLX-null mice under basal conditions. This data is now shown in the ***New Supplementary Figure 9***. Furthermore, we provide additional analysis of mass and CC3 staining of apoptosis at RT and thermoneutral 30°C that did not show any difference.

Core Body Temperature prior cold-exposure:

Cellular death assessment of BAT at RT (25°C) and Thermoneutral 30°C:

25°C

Apoptosis frequency

Thermoneutral 30°C

3. Since intracellular calcium molecules are dynamically cycling among cytosols, ER, and mitochondria, it is important to assess whether calcium dynamics are altered in the cytosol or ER in the siNCLX and NCLX KO cells. In addition, cytosolic calcium handling machineries, such as SERCA or RyR2, have been implicated in brown/beige fat thermogenesis (e.g., Ikeda et al., Nat Med 2018), thus, the authors also need to evaluate if these calcium handling systems are altered in the NCLX KO or KD cells

Response: We thank the reviewer for bringing up this interesting topic, we now provide new experiments in the revised version (please see the **New Supplementary figure 2** and pp.6). Interestingly, our results show that while Ca^{2+} entry to the cytosol was not altered upon NE-stimulation in the NCLX KO BA, Ca^{2+} clearance was significantly

impaired leading to increased duration in the cytosolic Ca^{2+} response. Therefore, our experiments demonstrate that impairing mitochondrial Ca^{2+} handling strongly impacts cytosolic Ca^{2+} clearance rate in the adrenergically activated NCLX KO BA. An increase in cytosolic Ca^{2+} response duration is consistent with the observed mitochondrial Ca^{2+} overload and in agreement with previous studies showing a role for NCLX in controlling cytosolic Ca^{2+} response (Ben-Kasus Nissim et al. 2017; Nita et al. 2014).

Following the reviewer's suggestion, we analyzed the content of central components involved in calcium cycling as suggested. We found that NCLX deletion did not affect the expression levels of PMCA, RyR or SERCA2; further supporting that changes in cytosolic Ca^{2+} are just reflecting the unique metabolic properties of activated NCLX KO BA.

Cytosolic Ca^{2+} responses measured by Fura-2AM in NCLX KO and WT BA:

4. In Figure 4, the authors spent almost whole figure (Fig. 4a-4g) to address the decrease of maximal VO_2 after acute cold or CL treatment was not due to change of UCP1 or other mitochondrial proteins. However, one wouldn't expect 6 hrs. of cold or CL treatment could lead to significant changes at the protein levels. What's important and yet missing in this figure was the assessment of changes in mitochondrial activity and/or changes in post-translational modifications of mitochondrial proteins. In Fig. 4h, in addition to the results of the quantification of spare respiration capacity, please also show the original seahorse curve.

Response: We thank the reviewer for helping us improving Figure 4. In the **New Figure 4**, we carried a detailed analysis of mitochondrial complexes activity (complex I, II and IV) before and after adrenergic stimulation, and found that under non-stimulated conditions Complex I activity was indeed reduced. However, on its own, reduction in Complex I activity was insufficient to lead to a bioenergetic phenotype as shown in measurements of the basal respiration of intact NCLX KO BA presented now in panel

4p. Furthermore, we also carried analysis of super-complex assembly on Blue-Native PAGE and our results show minimal and statistically insignificant changes in WT or NCLX KO BAT before/after stimulation. The original seahorse curve is now added in the **NEW Supplementary Figure 7**.

Activity of mitochondrial complexes:

Super-complex I+III assembly:

Basal Respiration:

Original respirometry curve of stimulated/non-stimulated WT and NCLX KO BA:

The data presented in Fig. 5d regarding mitochondrial swelling was not very convincing. EM images of mitochondrial ultrastructure would be an alternative approach to demonstrate morphological changes of mitochondria in the NCLX KO cells.

Response: Thank you for raising this important issue. Following the reviewer's suggestion, we have conducted EM analysis of stimulated BA of NCLX-null versus WT and now shown in the **NEW Figure 5** (panels e-h and pp.10). Our findings show a great damage to the cristae of the NE-stimulated NCLX-null versus WT BA mitochondria. This finding is fully consistent with our super-resolution fluorescence analysis on mitochondrial morphology presented in the same figure and subsequent supplementary figures (**New Supplementary Figure 8**) and the attached files of high-resolution images of stimulated NCLX-null and WT BA.

Moreover, to further link between mitochondrial Ca²⁺ and swelling, we carried-out new set of experiments presented in the **NEW Figure 5** (panels i-k and pp.10) showing that when extra or intracellular Ca²⁺ is chelated by EGTA and BATPA-AM respectively, or when the MCU is blocked and thereby preventing mitochondrial Ca²⁺ entry using the pharmacological inhibitor Ru360, NE-stimulated NCLX KO mitochondria remain intact

and swelling events are inhibited. Thus, this data strongly demonstrates that the swelling events observed in the stimulated NCLX KO BA are a result of mitochondrial Ca^{2+} overload.

Interestingly, a parallel study from the Mootha lab was published just recently focusing on the role of the mitochondrial Ca^{2+} uniporter (MCU) in BAT (Flicker et al. 2019). No phenotype was found in basal nor in cold-stimulated conditions for the BAT-MCU KO mice. That study demonstrates that the MCU is dispensable for BAT activity similar to what was shown in the heart (Luongo et al. 2015). We suggest that this further supports our study showing that while BAT function seems to be independent of mitochondrial Ca^{2+} uptake by MCU, once it is present in the mitochondrial matrix, Ca^{2+} levels must be tightly regulated by NCLX activity to prevent Ca^{2+} overload, activation of mPTP and mitochondrial swelling and cellular death as demonstrated in this Figure.

Electron Microscopy of stimulated NCLX KO and WT BA:

Ca²⁺ chelation and blocking its entry to the mitochondria prevent swelling in stimulated NCLX KO BA:

6. In Fig. 5a and 5b, the authors showed more cytochrome c release from mitochondria in KO cells using immunofluorescent staining method. These data need to be confirmed by Western blots using samples from subcellular fractionation.

Response: We thank the reviewer for raising this issue. Following much efforts, we

have to conclude that due to the low abundance of Cyt. c in brown adipocytes, WB-analysis of Cyt. c in these cells is technically not possible with current tools. We tried over months to establish WB-detection of Cyt. c release from NE-stimulated BA by WB. Despite detection of satisfactory levels of COXI and other mitochondrial proteins in the mitochondrial fractions in all samples (treated/ non-treated WT and KO), we were unable to detect Cyt. c using WB (please see the gel below with the subcellular fractionation). Apparently, yields of mitochondria from cultured primary BA may be too low for Cyt. c detection using cellular fractionation in WB. We tried several antibodies that were verified in other systems but without any result. Note that consistent with this technical hurdle, to our knowledge we found no other previous study that did such analysis from primary BA. Moreover, data from a recent study that analyzed mouse BAT proteomics is also in-line with this as ratio of Cyt c./VDAC (Porin) is ~10% (Kazak et al. 2017, S. Dataset1), while in liver for example, in genotype-matched mice and under chow-diet, a quantitative proteomics study showed a ratio of Cyt c./VDAC that is more than 400% (Lohr et al. 2016, Table S2) indicating that indeed the quantity of Cyt. c is not-abundant in BAT and might be hard to detect by WB analysis and subcellular fractionation.

7. Was there any change in tissue size or weight of BAT observed in the NCLX null mice given the fact that the KO mice displayed increased cell death in BAT (Fig. 5e-5h)? These data would be important for the interpretation of the data shown in Fig. 6f. If the KO mice had a smaller BAT mass, it would be conceivable to observe less FDG uptake in BAT region. If that was the case, it might be prudent to normalize the FDG uptake data to BAT mass

Response: Thank you for raising this important point. We provide in the **NEW Supplementary Figure 9** iBAT mass measurements of these mice and we found no significant difference in their mass at RT or thermoneutrality.

Our attached videos in the manuscript (**Supplementary videos 1-3**) show a 3D projection of maximum intensity projection (MIP) of the signal which could be confusing as it may imply that the iBAT mass was lower. We hope that this issue is clarified now.

8. In the discussion, the authors concluded that “Mechanistically, this is the first study that shows a physiological role of PKA-regulated mitochondrial Ca²⁺ extrusion via NCLX activation”. This was apparently overstated because as the authors noted in the Introduction that it has already been shown that PKA-mediated phosphorylation of NCLX rescued mitochondrial Ca²⁺efflux in depolarized neurons lacking PINK1.

Response: Thank you for pointing that our claim was not presented clearly enough. Indeed, the regulation of NCLX by PKA was demonstrated in the Kostic et al report. Following your note, we have revised the manuscript (pp.14). we meant to state that this is the first study showing a *physiological* process of activation for NCLX via a PKA-mediated pathway. The previous study was done in a Parkinson cellular model of PINK-1 deficient neurons and PKA activation was done artificially by introducing forskolin.

We thank the reviewer again for the constructive critique and guidance for improving our paper and its quality.

Reviewer #3 (Remarks to the Author):

In this manuscript, Assali et al. investigate the role of mitochondrial sodium/calcium exchanger (NCLX) in brown fat function, using cultured cells and mice. The authors show that loss of NCLX (1) leads to calcium accumulation in the mitochondria after adrenergic stimulation, (2) impairs cold-induced thermogenesis (3) causes cell death after adrenergic stimulation. The contribution of mitochondrial calcium signaling to brown fat physiology has not been well understood, and represents a topic of great interest to the fields of mitochondrial biology, calcium signaling and metabolisms. Even though the authors convincingly show that NCLX is important for non-shivering thermogenesis in mice, given that this is a whole body knockout, it's not possible to attribute the observed phenotype to brown fat-specific NCLX function. The authors show the importance of NCLX in protecting brown fat cells in response to adrenergic stimulation, but the finding that NCLX protects cells from PTP-mediated cell death is not novel and has been shown in the heart by Luongo et al (PMID: 28445457). In addition, PKA-regulation of NCLX function in brown adipocytes needs further experimental support.

Response: We thank the reviewer for a thorough examination and reading of our work. We think your constructive criticism greatly enhanced our experimental design and improved the impact and novelty of our manuscript.

Major points:

1. Loss of NCLX in adult mice in the heart was shown to be lethal (PMID: 28445457). It is very surprising the NCLX knockout animals survive. Western blot in Figure 3d shows that there is still detectable NCLX protein in the KO animals. This should be discussed in the manuscript.

Response: Thank you for bringing up this important issue. In brief, we think there might be compensatory mechanisms going in vital tissues like the heart. Luongo et al, who reported this heart phenotype, also claimed that mice born with the deletion of NCLX were completely viable and showed no phenotype, unlike the mice that lost NCLX during maturity. We believe that it is related to the fact that heart during the development can activate compensatory mechanisms, being more plastic, while adult hearts cannot. Please note, primary NCLX KO BA have no mitochondrial Ca^{2+} efflux at all as shown in **Figure 2**. In the discussion, we added a paragraph discussing this issue on pp.14.

Indeed, the WB shows a minor band but this is due to the quality of the antibody that genotyping of RNA-seq did not detect any NCLX transcript at all.

2. One of the main claims in this manuscript is PKA regulation of mitochondrial calcium efflux, via NCLX. However, no data that place NCLX and PKA on the same pathway are presented here. What happens to mitochondrial calcium efflux if PKA is activated in NCLX KO cells (by forskolin addition for example)? The authors refer to a previous paper by Dr. Sekler that shows phosphorylation of NCLX by PKA. The previous study was conducted in neurons, is the same signaling pathway conserved in brown adipocytes? Does PKA have NCLX-independent effects on mitochondrial calcium cycle in adipocytes? These questions should be addressed to strengthen the author's argument that the PKA regulates NCLX function in BAT.

Response: We thank the reviewer for this comment. In the revised manuscript, the entire **Figure 1** is dedicated to provide data that demonstrates the effect of NE on mitochondrial Ca^{2+} through PKA activity. This includes: 1) NE dependent activation of NCLX is fully blocked by the PKA blocker H-89 in brown adipocytes. 2) Ca^{2+} efflux is dependent on forskolin activity (PKA activator) when done in the presence of non-PKA-activating Ca^{2+} mobilizer (ATP). Thus, these data strongly support a link between PKA activity and NCLX function in BAT, as it was previously shown in a model of neurons. In contrast, please note that Ca^{2+} efflux in NCLX KO BA, as shown in various ways in **Figure 2 and NEW Supplementary Figure 1**, is still blocked under physiological PKA activating conditions triggered through NE-stimulation.

3. Ikeda et al. showed that calcium cycling is important in beige fat for thermogenesis, independent of UPC-1 (PMID: 29131156). It is reasonable to think that NCLX may be important for calcium cycling in this tissue. The thermogenesis phenotype observed in NCLX KO mice used in this study could be a combination of brown and beige fat phenotypes, and not only due to brown fat. In addition, Jackson Labs reposts that the NCLX KO mouse strain used in this study has "decreased circulating glucose level" (mouse strain 026242). These are all confounding factors while determining the BAT-specific effects of NCLX loss. Using a conditional NCLX mouse model may be beyond the scope of this study, but all these caveats should be addressed in the discussion.

Response: Thank you for all of these important comments and insights. We now revised the discussion according to the reviewer's suggestions (please see pp.14).

Moreover, we have related to the contribution of NCLX to global intracellular Ca^{2+} in our new data shown in **Supplementary Figure 2** and pp.6.

4. I strongly disagree with the author's conclusion that BAT dysfunction should be the main reason for the observed phenotype. β -3 stimulation experiments show a defect in BAT, but do not exclude defects in other tissues.

Response: We understand the concern of the reviewer and therefore we revised the

conclusion accordingly and we now write that β -3 stimulation experiments show “*defective non-shivering thermogenesis*” in NCLX KO mice, instead of “*BAT defective thermogenesis*”. This claim was also revised in the results section related to Figure 3, including the title of this figure (pp.8).

Minor points:

1. The manuscript will benefit from editing for spelling and grammar. Examples: However, for TCA cycle to meet up with the energy demand... should be: However, for TCA cycle to meet/ keep up with the energy demand. **Response:** corrected.

A recent study showed that that PKA-mediated...: delete “that” **Response:** corrected. There are many others.

Response: We thank the reviewer for the meticulous reading and we now fixed all of these and other mistakes.

2. The figures should also be edited, especially the fonts. Some fonts are stretched out, font sizes and scale bars are variable in figures. In addition, different scale bars are used for similar experiments (monitoring mitochondrial calcium for example, in figures 1a and 1b), making it difficult to compare the data.

Response: We apologize for the formatting problems, in the revised manuscript we aligned all the fonts and scale bars of Ca^{2+} in the different figures.

3. NMDG+ treatment of cells is likely to affect a lot of different cellular processes. The authors should use NCLX inhibitor CPG37157 in experiments shown in Fig1 to assess the effects of NCLX inhibition more specifically.

Response: CGP37157 indeed inhibits mitochondrial Ca^{2+} efflux, however, it inhibits L-type Ca^{2+} channels almost as potently if not more (Ruiz et al. 2014; Thu et al. 2006), therefore its usage is not favorable. The NMDG+ experiments were used just as a paradigm showing that Ca^{2+} efflux is Na^{+} -dependent and to recapitulate the studies done by Cannon and Nedegaard in the 70s (Al-Shaikhaly et al. 1979), where they showed that Na^{+} -dependent mitochondrial Ca^{2+} is essential for thermogenesis in BAT (however the molecular identity of NCLX was unknown at that time).

4. In the discussion the authors state “Remarkably, this study shows that mitochondrial Ca^{2+} overload is not deleterious per se for mitochondrial bioenergetics as long as the mPTP remains closed”. I agree that this would be an interesting finding, however, the authors do not show that there’s still mitochondrial calcium overload after NIM811 treatment of mice. The authors should look at mitochondrial swelling and calcium overload after NIM811 treatment.

Response: We thank the reviewer for these important suggestions. Indeed, we think this is a novel finding in our manuscript and in the revised version we carried extensive studies to elaborate all of the reviewer's inquiries, now shown in the **New Revised Figure 6**. Our new results show prevention of mitochondrial swelling and maintenance of cellular viability despite the mitochondrial Ca^{2+} overload in the NCLX KO cells pretreated with NIM811 (panels 6c-h, pp.12). We believe these new pieces of data strengthen our claim further.

Mitochondrial Ca^{2+} assessment in BA pretreated with NIM811 or vehicle:

Swelling analysis in BA pretreated with NIM811 or vehicle:

We thank the reviewer again for his/her helpful criticism that guided us to greater insights.

References

- Al-Shaikhaly, M H, J Nedergaard, and B Cannon. 1979. "Sodium-Induced Calcium Release from Mitochondria in Brown Adipose Tissue." *Proceedings of the National Academy of Sciences of the United States of America* 76 (5): 2350–53.
- Ben-Kasus Nissim, Tsipi, Xuexin Zhang, Assaf Elazar, Soumitra Roy, Judith A Stolwijk, Yandong Zhou, Rajender K Motiani, et al. 2017. "Mitochondria Control Store-Operated Ca(2+) Entry through Na(+) and Redox Signals." *The EMBO Journal* 36 (6): 797–815. <https://doi.org/10.15252/embj.201592481>.
- Flicker, Daniel, Yasemin Sancak, Eran Mick, Olga Goldberger, and Vamsi K Mootha. 2019. "Exploring the In Vivo Role of the Mitochondrial Calcium Uniporter in Brown Fat Bioenergetics." *Cell Reports* 27 (5): 1364–1375.e5. <https://doi.org/10.1016/j.celrep.2019.04.013>.
- Jiang, Lin-Hua, Fatema Mousawi, Xuebin Yang, and Sebastien Roger. 2017. "ATP-Induced Ca(2+)-Signalling Mechanisms in the Regulation of Mesenchymal Stem Cell Migration." *Cellular and Molecular Life Sciences : CMLS* 74 (20): 3697–3710. <https://doi.org/10.1007/s00018-017-2545-6>.
- Kazak, Lawrence, Edward T Chouchani, Irina G Stavrovskaya, Gina Z Lu, Mark P Jedrychowski, Daniel F Egan, Manju Kumari, et al. 2017. "UCP1 Deficiency Causes Brown Fat Respiratory Chain Depletion and Sensitizes Mitochondria to Calcium Overload-Induced Dysfunction." *Proceedings of the National Academy of Sciences of the United States of America* 114 (30): 7981–86. <https://doi.org/10.1073/pnas.1705406114>.
- Kostic, Marko, Tomer Katoshevski, and Israel Sekler. 2018. "Allosteric Regulation of NCLX by Mitochondrial Membrane Potential Links the Metabolic State and Ca(2+) Signaling in Mitochondria." *Cell Reports* 25 (12): 3465–3475.e4. <https://doi.org/10.1016/j.celrep.2018.11.084>.
- Kostic, Marko, Marthe H R Ludtmann, Hilmar Bading, Michal Hershfinkel, Erin Steer, Charleen T Chu, Andrey Y Abramov, and Israel Sekler. 2015. "PKA Phosphorylation of NCLX Reverses Mitochondrial Calcium Overload and Depolarization, Promoting Survival of PINK1-Deficient Dopaminergic Neurons." *Cell Reports* 13 (2): 376–86. <https://doi.org/10.1016/j.celrep.2015.08.079>.
- Lohr, Kerstin, Fiona Pacht, Amin Moghaddas Gholami, Kerstin E Geillinger, Hannelore Daniel, Bernhard Kuster, and Martin Klingenspor. 2016. "Reduced Mitochondrial Mass and Function Add to Age-Related Susceptibility toward Diet-Induced Fatty Liver in C57BL/6J Mice." *Physiological Reports* 4 (19). <https://doi.org/10.14814/phy2.12988>.
- Luong, Timothy S, Jonathan P Lambert, Ancai Yuan, Xueqian Zhang, Polina Gross, Jianliang Song, Santhanam Shanmughapriya, et al. 2015. "The Mitochondrial Calcium Uniporter Matches Energetic Supply with Cardiac Workload during Stress and Modulates Permeability Transition." *Cell Reports* 12 (1): 23–34. <https://doi.org/10.1016/j.celrep.2015.06.017>.
- Nita, Iulia I, Michal Hershfinkel, Chase Kantor, Guy A Rutter, Eli C Lewis, and Israel Sekler. 2014. "Pancreatic Beta-Cell Na+ Channels Control Global Ca2+ Signaling and Oxidative Metabolism by Inducing Na+ and Ca2+ Responses That Are Propagated into Mitochondria." *FASEB Journal : Official Publication of the Federation of American Societies for Experimental Biology* 28 (8): 3301–12. <https://doi.org/10.1096/fj.13-248161>.
- Palty, Raz, Ehud Ohana, Michal Hershfinkel, Micha Volokita, Vered Elgazar, Ofer Beharier, William F Silverman, Miriam Argaman, and Israel Sekler. 2004. "Lithium-Calcium Exchange Is Mediated by a Distinct Potassium-Independent Sodium-Calcium Exchanger." *The Journal of Biological Chemistry* 279 (24): 25234–40. <https://doi.org/10.1074/jbc.M401229200>.
- Ruiz, A, E Alberdi, and C Matute. 2014. "CGP37157, an Inhibitor of the Mitochondrial Na+/Ca2+ Exchanger, Protects Neurons from Excitotoxicity by Blocking Voltage-Gated Ca2+ Channels." *Cell Death & Disease* 5 (April): e1156. <https://doi.org/10.1038/cddis.2014.134>.
- Thu, Le Thi, Joung Real Ahn, and Sun-Hee Woo. 2006. "Inhibition of L-Type Ca2+ Channel by Mitochondrial Na+-Ca2+ Exchange Inhibitor CGP-37157 in Rat Atrial Myocytes." *European Journal of Pharmacology* 552 (1–3): 15–19. <https://doi.org/10.1016/j.ejphar.2006.09.013>.

REVIEWERS' COMMENTS:

Reviewer #1 (Remarks to the Author):

I have carefully examined the revised manuscript and author's rebuttal. The authors adequately responded to the critique by performing a number of new experiments and modifying the text of the paper. The manuscript is now suitable for publication.

Reviewer #2 (Remarks to the Author):

Overall, the authors were very responsive to previous comments, and the additional experiments and discussions have largely strengthened the revised manuscript. There are a couple of points that need to be addressed before it can be considered for publication in Nature Communications.

1. The major caveat of using the whole-body NCLX KO mice is that it is difficult to pinpoint which tissue is responsible for the observed phenotypes. While the authors recognized this pitfall, they still solely emphasized on the role of BAT in mediating the impaired thermogenic phenotype of the KO mice. To avoid misleading the readers, I would suggest the authors to amend the main text (e.g., last paragraph of the introduction, results and discussions) by recognizing the potential caveat of the whole-body KO mice, and acknowledging the phenotypes of the KO mice could be contributed from other tissues, such as skeletal muscle, heart, and others. While making tissue-specific NCLX KO mice is beyond the scope of the current study, it certainly warrants future investigation.
2. To be consistent with the y-axis of quantified bar graphs in Fig 4p and 4q, please change the OCR (%) to the real value of OCR (pmoles O₂/min/1000 cells) in the y-axis of NEW Supplementary Figure 7.

Reviewer #3 (Remarks to the Author):

The revised manuscript is much improved. The authors addressed all of concerns. Their data support the author's claims. The experiments are well controlled. The findings presented in this manuscript provide interesting insights on the role of calcium cycling in cellular and organismal bioenergetics.

REVIEWERS' COMMENTS:

Reviewer #1 (Remarks to the Author):

I have carefully examined the revised manuscript and author's rebuttal. The authors adequately responded to the critique by performing a number of new experiments and modifying the text of the paper. The manuscript is now suitable for publication.

Response: We greatly appreciate the reviewer's positive feedback and their efforts in improving the manuscript.

Reviewer #2 (Remarks to the Author):

Overall, the authors were very responsive to previous comments, and the additional experiments and discussions have largely strengthened the revised manuscript. There are a couple of points that need to be addressed before it can be considered for publication in Nature Communications.

Response: We thank the reviewer for their comment and support of our manuscript.

1. The major caveat of using the whole-body NCLX KO mice is that it is difficult to pinpoint which tissue is responsible for the observed phenotypes. While the authors recognized this pitfall, they still solely emphasized on the role of BAT in mediating the impaired thermogenic phenotype of the KO mice. To avoid misleading the readers, I would suggest the authors to amend the main text (e.g., last paragraph of the introduction, results and discussions) by recognizing the potential caveat of the whole-body KO mice, and acknowledging the phenotypes of the KO mice could be contributed from other tissues, such as skeletal muscle, heart, and others. While making tissue-specific NCLX KO mice is beyond the scope of the current study, it certainly warrants future investigation.

Response: We again address this point and the pitfall of using a global NCLX KO mouse model throughout the whole manuscript (e.g. the results and discussion sections) as suggested by the reviewer.

2. To be consistent with the y-axis of quantified bar graphs in Fig 4p and 4q, please change the OCR (%) to the real value of OCR (pmoles O₂/min/1000 cells) in the y-axis of NEW Supplementary Figure 7.

Response: We thank the reviewer for raising this point. The data in supplementary figure 7 is now presented in (pmoles O₂/min/1000 cells) as suggested.

Reviewer #3 (Remarks to the Author):

The revised manuscript is much improved. The authors addressed all of concerns. Their data support the author's claims. The experiments are well controlled. The findings presented in this manuscript provide interesting insights on the role of calcium cycling in cellular and organismal bioenergetics.

Response: We thank the reviewer for his/her compliments and praise on our revised manuscript and their devoted time in improving our manuscript.